# Eicosanoid and eicosanoid-related inflammatory mediators and exercise intolerance in heart failure with preserved ejection fraction

Emily S. Lau [1,2,16], Athar Roshandelpoor[3,16], Shahrooz Zarbafian[1,4], Dongyu Wang[3,5], James S. Guseh [1,2], Norrina Allen[6], Vinithra Varadarajan[7], Matthew Nayor[8], Ravi V. Shah[9], Joao A. C. Lima [7], Sanjiv J. Shah [10,11], Bing Yu [12], Mona Alotaibi[13], Susan Cheng [14], Mohit Jain[15], Gregory D. Lewis[1,2] & Jennifer E. Ho [2] ✉

Systemic inflammation has been implicated in the pathobiology of heart failure with preserved ejection fraction (HFpEF). Here, we examine the association of upstream mediators of inflammation as ascertained by fatty-acid derived eicosanoid and eicosanoid-related metabolites with HFpEF status and exercise manifestations of HFpEF. Among 510 participants with chronic dyspnea and preserved LVEF who underwent invasive cardiopulmonary exercise testing, we find that 70 of 890 eicosanoid and related metabolites are associated with HFpEF status, including 17 named and 53 putative eicosanoids (FDR $q$-value < 0.1). Prostaglandin (15R-PGF2α, 11ß-dhk-PGF2α) and linoleic acid derivatives (12,13 EpOME) are associated with greater odds of HFpEF, while epoxides (8(9)-EpETE), docosanoids (13,14-DiHDPA), and oxylipins (12-OPDA) are associated with lower odds of HFpEF. Among 70 metabolites, 18 are associated with future development of heart failure in the community. Pro- and anti-inflammatory eicosanoid and related metabolites may contribute to the pathogenesis of HFpEF and serve as potential targets for intervention.

Heart failure is a major public health concern, affecting over 6 million individuals and accounting for more than 1 million hospital admissions per year in the United States[1]. HF with preserved ejection fraction (HFpEF) has emerged as the leading form of HF, yet causal and contributing factors to HFpEF are poorly characterized[2]. While the hallmark feature of HFpEF is exercise intolerance, HFpEF is a highly heterogeneous condition characterized by deficits across multiple organ systems beyond the cardiovascular system itself[3–5]. Recent evidence has implicated systemic inflammation in the pathogenesis of HFpEF and has been hypothesized to mediate its association between comorbidities such as obesity, diabetes mellitus (DM), and hypertension (HTN)[6,7]. Previous studies have demonstrated that circulating

markers of systemic inflammation including C-reactive protein (CRP) are associated with incident HFpEF[8–10]. However, specific inflammatory pathways involved in the development of HFpEF remain unclear.

Upstream activation of systemic inflammation in humans is governed in part by a class of small bioactive lipids enzymatically derived from polyunsaturated fatty acids, termed eicosanoids[11–13]. Eicosanoid and eicosanoid-related metabolites, including thromboxanes, prostaglandins, leukotrienes, and resolvins, regulate the upstream initiation of systemic inflammation and have previously been related to fundamental physiologic functions including the modulation of renal vascular tone, peripheral resistance, and endothelial function[14–16]. Until recently, quantification of eicosanoids and related metabolites has

been limited in scale, precluding detailed understanding of the role of upstream inflammatory mediators in HFpEF pathobiology. However, the introduction of mass-spectrometry (MS)-based analytics has enabled rapid and sensitive quantification of eicosanoids and eicosanoid-related metabolites in human plasma at large-scale[17]. Early investigations of eicosanoids and related metabolites leveraging MS-based approaches have demonstrated strong associations between plasma eicosanoids with blood pressure and DM, both clinical precursors to HFpEF[18,19].

In this context, we performed large-scale comprehensive profiling of >500 pro- and anti-inflammatory eicosanoid and eicosanoid-related metabolites using liquid chromatography-MS (LC-MS) to better understand upstream inflammatory mediators as determinants of HFpEF. We leveraged a hospital-based cohort of individuals who underwent detailed physiologic phenotyping with invasive cardiopulmonary exercise testing to examine eicosanoid profiles of HFpEF and gain a deeper understanding of specific exercise manifestations of HFpEF across organ systems. Lastly, to corroborate findings across a separate cohort, we examined top eicosanoid and eicosanoid-related metabolites in the Multi-Ethnic Study of Atherosclerosis (MESA) study, a longitudinal community-based sample with incident HF outcomes.

## Results

Among 510 participants in MGH CPET (age 56 ± 16 years, 63% women), 257 had evidence of physiologic HFpEF. Clinical and exercise characteristics by HFpEF status and sex are displayed in Table 1 and Supplementary Data 1–3. Compared with individuals without HFpEF, participants with HFpEF were older (mean age 61 vs 50 years) and had greater burden of comorbidities including obesity (7% vs 6%), HTN (64% vs 40%), DM (23% vs 10%), and AF (20% vs 6%). Use of medications including aspirin, statins, and diuretics was higher in participants with vs without HFpEF (aspirin: 43% vs 28%, statin: 48% vs 22%, diuretics: 41% vs 13%). Renal function was lower in individuals with HFpEF (creatinine: 1.08 mg/dL vs 0.92 mg/dL, estimated glomerular filtration rate: 72 vs 83 mL/min/1.73m²). Traditional cardiovascular biomarkers showed higher NT-proBNP (106 vs 43 pg/mL) and hsCRP (2.84 mg/L vs 1.16 mg/L) among individuals with vs without HFpEF. With respect to exercise characteristics, patients with HFpEF had lower peak $VO_2$ (14.7 mL/kg/min vs 19.7 mL/kg/min) at similar RER.

### Association of eicosanoids and eicosanoid-related metabolites with HFpEF

Of 890 eicosanoid and eicosanoid-related metabolites assayed, 70 (8%) were associated with HFpEF status in the MGH CPET cohort in primary analyses (FDR $q$-value < 0.10, Fig. 1). Results for eicosanoid and related metabolites with known molecular identity are shown in Table 2, with full results including putative eicosanoids and eicosanoid-related metabolites displayed in Supplementary Data 4. We found that 21 eicosanoids and eicosanoid-related metabolites (5 named and 16 putative) were associated with higher odds of HFpEF, including prostaglandin, linoleic acid, and hydroxyeicosatrienoic acid derivatives. The metabolites with the largest observed magnitude of effect were prostaglandins, including 15R-prostagladin F2α (PGF2α) and 11ß-dihydro-15-keto PGF2α (11ß-dhk-PGF2α). Specifically, a 1-standard deviation (SD) higher 15R-PGF2α and 11ß-dhk-PGF2a were associated with >1.5-fold increased odds of having HFpEF (odds ratio [OR] = 1.70, 95% CI [confidence interval] = 1.30–2.26, $p$ = 0.0002 and OR = 1.55, 95% CI = 1.22–2.03, $p$ = 0.0007, respectively). By contrast, 49 eicosanoids (12 named, 37 putative eicosanoids) were associated with lower odds of HFpEF. This included specific epoxides, oxylipins, docosahexaenoic acid (DHA) derivatives, prostaglandin F2α, and resolvins. The largest magnitude of effect was observed for 8(9)-epoxy-5Z,11Z,14Z,17Z-eicosatetraenoic acid (EpETE), an epoxide (OR = 0.58, 95% CI = 0.45–0.73, $p$ = 0.004) and 13,14-dihydroxy-4Z,7Z,10Z,13Z,16Z-

docosapentaenoic acid (DiHDPA), a DHA metabolite (OR = 0.62, 95% CI = 0.48–0.79, $p$ = 0.0002). Of note, we found 12,13 diHOME, a known exercise lipokine, was associated with lower odds of HFpEF (OR = 0.66, 95% CI = 0.53–0.83, $p$ = 0.0003). Resolvin D1, a member of the resolvin family of metabolites involved in resolution of inflammation, was also associated with lower odds of HFpEF (OR = 0.74, 95% CI = 0.59–0.91).

**Table 1 | Baseline Clinical and CPET Characteristics by HFpEF Status in the MGH CPET Sample**

| N | No HFpEF N = 253 | HFpEF N = 257 | p-value |
|---|---|---|---|
| **Clinical characteristics** | | | |
| Age, years | 50 (16) | 61 (13) | 8.21 E−15 |
| Women, n (%) | 166 (66%) | 154 (60%) | 0.22 |
| White race, n (%) | 240 (95%) | 246 (96%) | 0.22 |
| BMI, kg/m² | 27.4 (5.9) | 31.0 (6.6) | 9.33 E−11 |
| Obesity, n (%) | 76 (30%) | 137 (53%) | 5.77 E−08 |
| Hypertension, n (%) | 101 (40%) | 164 (64%) | 1.09 E−07 |
| Myocardial infarction, n (%) | 7 (3%) | 11 (4%) | 0.49 |
| Diabetes mellitus, n (%) | 24 (10%) | 58 (23%) | 9.59 E−05 |
| Atrial fibrillation, n (%) | 16 (6%) | 52 (20%) | 7.13 E−06 |
| Left atrial enlargement, n (%) | 38 (20%) | 88 (41%) | 9.37 E−06 |
| Left ventricular hypertrophy, n (%) | 22 (11%) | 62 (29%) | 2.94 E−05 |
| COPD, n (%) | 14 (5%) | 30 (12%) | 0.02 |
| ILD, n (%) | 14 (5%) | 16 (6%) | 0.89 |
| OSA, n (%) | 41 (16%) | 67 (26%) | 0.009 |
| Past or current smoking, n (%) | 78 (31%) | 123 (48%) | 0.00012 |
| Current smoking, n (%) | 10 (4%) | 7 (3%) | 0.60 |
| Diuretic use, n (%) | 34 (13%) | 106 (41%) | 4.03 E−12 |
| Aspirin use, n (%) | 70 (28%) | 109 (43%) | 0.001 |
| Statin use, n (%) | 56 (22%) | 123 (48%) | 1.59 E−09 |
| Creatinine (mg/dL) | 0.92 (0.23) | 1.08 (0.86) | 0.01 |
| Estimated GFR, mL/min/1.73m² | 83 (20) | 72 (22) | 1.20 E−07 |
| NT-pro BNP, pg/mL | 43.0 (23.5–104.4) | 105.8 (51.2–302.7) | 1.69 E−14 |
| hsCRP, mg/L | 1.16 (0.52–3.17) | 2.84 (1.24–5.98) | 6.46 E−12 |
| LV ejection fraction, % | 65 (6) | 65 (8) | 0.72 |
| **CPET characteristics** | | | |
| Peak VO₂, mL/kg/min | 19.7 (6.1) | 14.7 (3.9) | 2.22 E−16 |
| Predicted VO₂, % | 80 (18) | 73 (15) | 2.37 E−06 |
| Peak RER | 1.18 (0.11) | 1.15 (0.11) | 0.006 |
| VE/VCO₂ slope | 32.1 (5.7) | 34.1 (6.4) | 0.0002 |
| C(a-v)O₂, mL/100 mL | 11.0 (2.0) | 11.5 (2.2) | 0.009 |
| ΔSBP, mmHg | 49 (30) | 51 (37) | 0.44 |
| ΔPCWP/ΔCO, mmHg/L/min | 1.23 (0.54) | 3.39 (2.55) | 0.001 |
| ΔPAP/ΔCO, mmHg/L/min | 2.15 (1.98) | 4.15 (2.91) | 2.20 E−16 |
| Predicted heart rate, % | 88 (11) | 81 (15) | <0.001 |

Values are means (standard deviations) or median (inter-quartile range) unless otherwise noted. $p$ < 0.05 considered significant by t-test, chi-squared test, or Kruskal-Wallis (for NT-pro BNP and hsCRP).
*BMI* body mass index, *CO* cardiac output, *COPD* chronic obstructive pulmonary disease, *GFR* glomerular filtration rate, *hsCRP* high sensitivity C-reactive protein, *ILD* interstitial lung disease, *LV* left ventricular, *NT-pro BNP* N-terminal pro B-type natriuretic peptide, *OSA* obstructive sleep apnea, *PAP* pulmonary artery pressure, *PCWP* pulmonary capillary wedge pressure, *RER* respiratory exchange ratio, *SBP* systolic blood pressure.

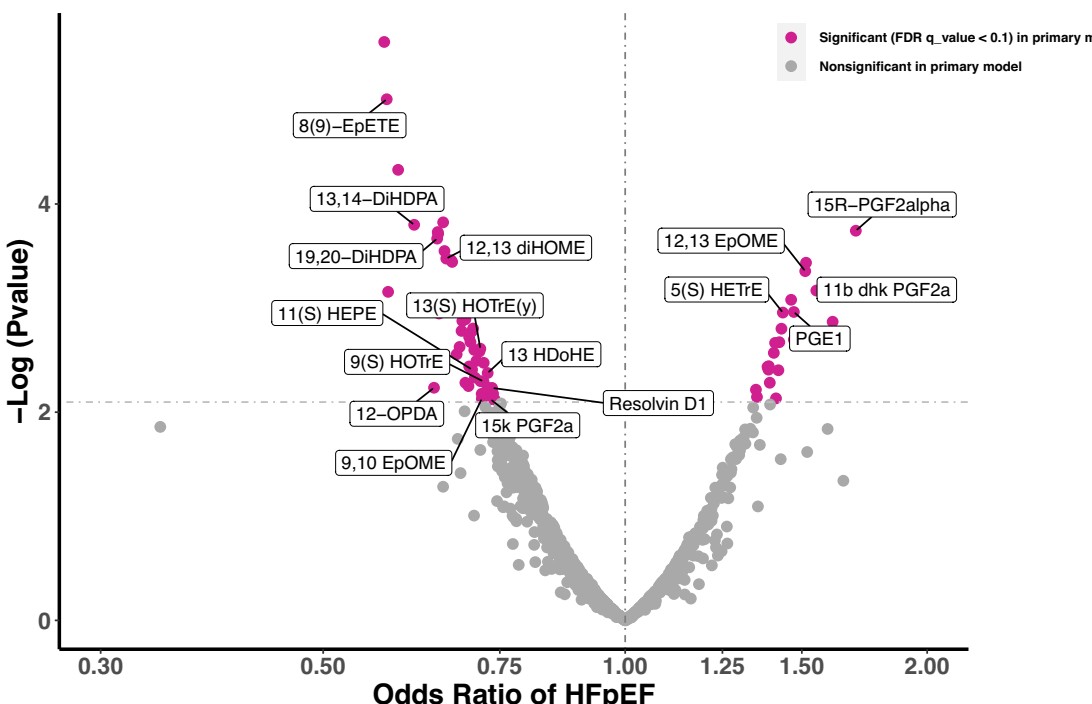

**Fig. 1 | Volcano plot for associations of eicosanoid and eicosanoid-related metabolites with HFpEF status in the MGH CPET sample.** A significant association was observed for 70 of 890 eicosanoids and eicosanoid-related metabolites in the primary model (adjusted for age, sex, and plate number, displayed in pink). Nonsignificant associations are denoted in gray (primary model). The dotted line denotes the level of significance ($p$-value < 0.008 equivalent to FDR $q$-value < 0.10 based on logistic regression model accounting for multiple hypothesis testing). HFpEF heart failure with preserved ejection fraction. Source data are provided as a Source Data file.

**Table 2 | Eicosanoid and eicosanoid-related metabolites significantly associated with HFpEF status**

| *m/z* | RT | Class | Putative ID | Primary model | | Secondary model | |
|---|---|---|---|---|---|---|---|
| | | | | OR (95% CI) | *p*-value | OR (95% CI) | *p*-value |
| **Associated with greater odds of HFpEF** | | | | | | | |
| 353.2330 | 2.0597 | Prostaglandin | 15R-PGF2α | 1.70 (1.30–2.26) | 0.0002 | 1.47 (1.11–1.97) | 0.008 |
| 353.2336 | 2.3865 | Prostaglandin | 11ß-dhk-PGF2α | 1.55 (1.22–2.03) | 0.0007 | 1.41 (1.10–1.84) | 0.009 |
| 355.2489 | 5.2293 | Eicosanoid | 12,13 EpOME | 1.51 (1.21–1.92) | 0.0004 | 1.48 (1.17–1.90) | 0.001 |
| 353.2336 | 2.2385 | Prostaglandin like | PGE1 | 1.47 (1.18–1.87) | 0.001 | 1.28 (1.02–1.63) | 0.04 |
| 381.2635 | 6.0865 | Eicosanoid | 5(S) HETrE | 1.44 (1.16–1.79) | 0.001 | 1.41 (1.13–1.78) | 0.003 |
| **Associated with lower odds of HFpEF** | | | | | | | |
| 377.2305 | 4.8408 | Epoxide | 8(9)-EpETE | 0.58 (0.45–0.73) | 0.00001 | 0.71 (0.55–0.92) | 0.01 |
| 361.2386 | 3.6877 | Eicosanoid | 13,14-DiHDPA | 0.62 (0.48–0.79) | 0.0002 | 0.68 (0.52–0.88) | 0.005 |
| 291.1953 | 3.3485 | Oxylipin | 12-OPDA | 0.64 (0.46–0.87) | 0.006 | 0.68 (0.49–0.93) | 0.02 |
| 361.2396 | 3.4903 | Docosanoid | 19,20-DiHDPA | 0.65 (0.51–0.81) | 0.0002 | 0.77 (0.60–0.98) | 0.03 |
| 313.2388 | 3.4472 | Eicosanoid | 12,13 diHOME | 0.66 (0.53–0.83) | 0.0003 | 0.80 (0.63–1.02) | 0.07 |
| 317.2123 | 4.0022 | Eicosanoid | 11(S) HEPE | 0.70 (0.55–0.88) | 0.004 | 0.78 (0.61–0.98) | 0.04 |
| 293.2123 | 4.1502 | Eicosanoid | 13(S) HOTrE(y) | 0.72 (0.58–0.89) | 0.002 | 0.83 (0.66–1.04) | 0.10 |
| 295.2280 | 5.2910 | Eicosanoid | 9,10 EpOME | 0.72 (0.56–0.91) | 0.007 | 0.76 (0.59–0.97) | 0.03 |
| 293.2122 | 3.9713 | Eicosanoid | 9(S) HOTrE | 0.72 (0.57–0.90) | 0.005 | 0.82 (0.64–1.05) | 0.11 |
| 343.2279 | 4.7298 | Eicosanoid | 13 HDoHE | 0.73 (0.58–0.90) | 0.004 | 0.81 (0.64–1.02) | 0.07 |
| 351.2182 | 1.4122 | Prostaglandin | 15k-PGF2a | 0.73 (0.58–0.92) | 0.008 | 0.75 (0.59–0.95) | 0.02 |
| 375.2177 | 2.4050 | Resolvin | Resolvin D1 | 0.74 (0.59–0.91) | 0.006 | 0.75 (0.60–0.95) | 0.02 |

Eicosanoids metabolites displayed met FDR *q*-value threshold <0.1 in the primary logistic regression model accounting for multiple hypothesis testing.
OR = odds of HFpEF status per 1-SD increase in EIC metabolite concentration.
Primary model adjusts for age, sex, and plate number.
Secondary model adjusts for age, sex, plate number, body mass index, diabetes, hypertension, present smoking, and prevalent MI.
*CI* confidence interval, *dhk-PGF2α* dihydro-15-keto-prostaglandin F2α, *DiHDPA* dihydroxy-4Z,7Z,10Z,13Z,16Z-docosapentaenoic acid, *diHOME* dihydroxy-9Z-octadecenoic acid, *EpETE* epoxy-5Z,11Z,14Z,17Z-eicosatetraenoic acid, *EpOME* epoxy-9Z-octadecenoic acid, *HDoHE* hydroxy-4Z,7Z,10Z,13Z,15E,19Z-docosahexaenoic acid, *HEPE* hydroxy-5Z,8Z,12E,14Z,17Z-eicosapentaenoic acid, *HETrE* hydroxy-6E,8Z,11Z-eicosatrienoic acid, *HOTrE* hydroxy-10E,12Z,15Z-octadecatrienoic acid,11E-octadecadienoic acid, *m/z* mass to charge ratio, *RT* retention time, *OPDA* oxo phytodienoic acid, *OR* odds ratio, *PGF2α* prostaglandin F2α.

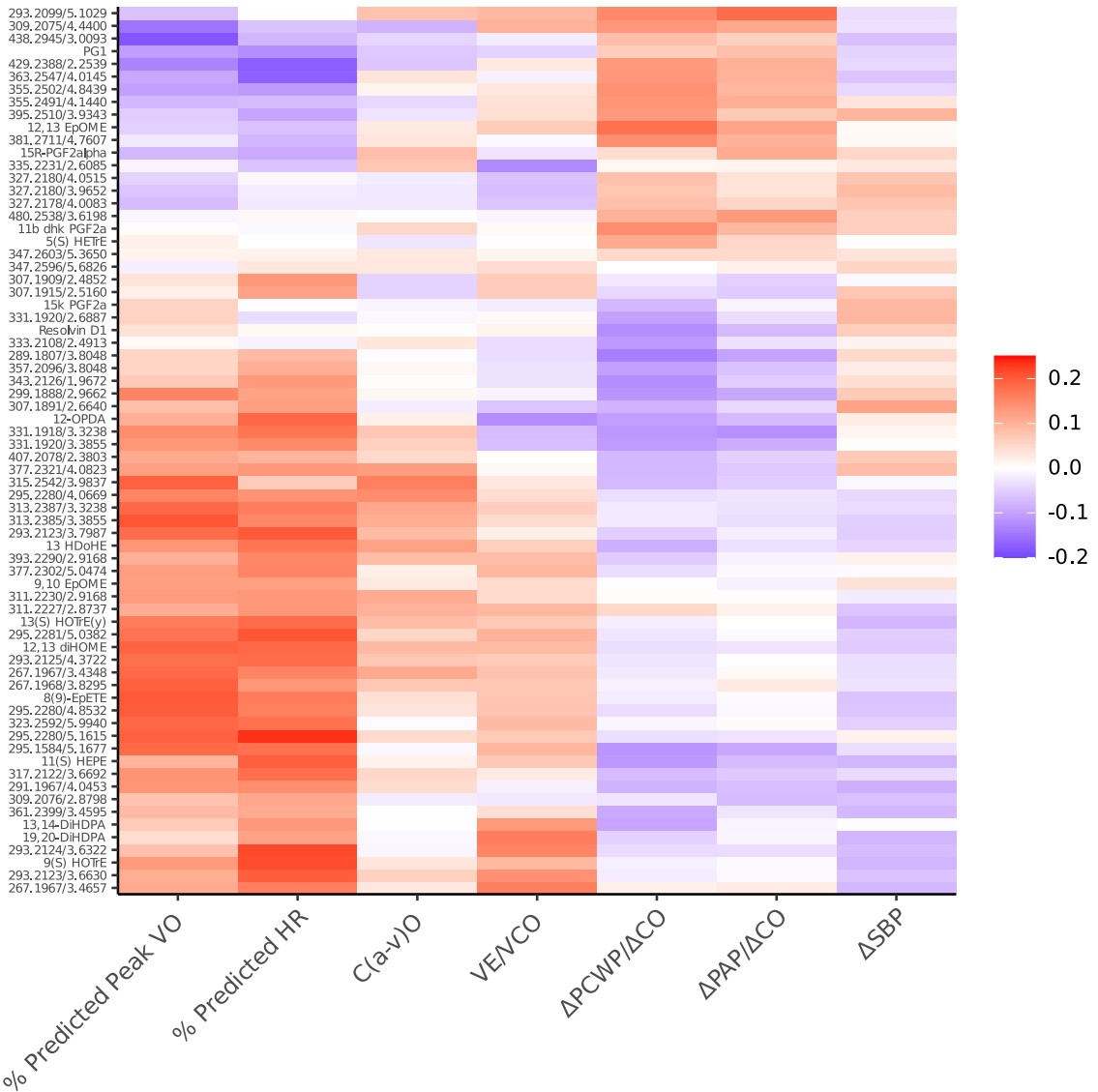

**Fig. 2 | Heatmap for associations of the 70 HFpEF-related eicosanoid and eicosanoid-related metabolites with exercise traits in the MGH CPET sample.** Color coding represents standardized ß-coefficient in primary model (X-SD change in exercise trait per 1-SD change in eicosanoid metabolite). Primary analyses are adjusted for age, sex, and plate number. Clustering is based on exercise traits (columns). Color scale indicates positive associations in red, and negative associations in blue. CO cardiac output, HR heart rate, PAP pulmonary artery pressure, PCWP pulmonary capillary wedge pressure, SBP systolic blood pressure. Source data are provided as a Source Data file.

## HFpEF-associated eicosanoids and eicosanoid-related metabolites display distinct associations with exercise responses

For the eicosanoids and related metabolites that were associated with HFpEF status, we separately examined their associations with overall exercise capacity (% predicted peak $VO_2$) and organ-specific exercise responses, including chronotropic response [% predicted HR], diastolic reserve [ΔPCWP/ΔCO slope]), ventilatory efficiency [VE/VCO$_2$ slope], peripheral oxygen extraction [C(a-v)$O_2$], blood pressure response [ΔSBP], and pulmonary vascular response [ΔPAP/ΔCO slope]. Figure 2 and Supplementary Fig. 1 display the heatmap of regression coefficients for the 70 HFpEF-related metabolites (rows) and their associations with exercise traits (columns). We clustered exercise traits into 3 groups using hierarchical clustering of eicosanoid profiles: (1) % predicted peak $VO_2$ and chronotropic response, (2) VE/VCO$_2$ and C(a-v)$O_2$, and (3) BP response, pulmonary vascular and diastolic reserve. Full results of HFpEF eicosanoid and related metabolite associations with exercise traits are shown in Supplementary Data 5. Of the 70 HFpEF-related eicosanoid and related metabolites, 27 (38%) were significantly associated with % predicted peak $VO_2$, 43 (61%) with chronotropic response (% predicted heart rate), 4 (6%) with diastolic reserve (ΔPCWP/ΔCO), 1 (1%) with pulmonary vascular reserve (ΔPAP/ΔCO), 3 (4%) with peripheral oxygen extraction (C[a-v]$O_2$), and 6 (8%) with ventilatory efficiency (VE/VCO$_2$). There were no eicosanoids or eicosanoid-related metabolites associated with BP response. We observed generally consistent directionality (eicosanoids associated with greater odds of HFpEF were also associated with worse exercise traits). Of note, there was considerable heterogeneity across eicosanoid and related metabolite associations with exercise traits. For example, the epoxide associated with lower odds of HFpEF (8(9)-EpETE) was associated with higher % predicted peak $VO_2$ ($p = 0.005$) and better chronotropic response ($p = 0.0009$), but no other exercise traits.

We performed PCA to understand the association of potentially correlated eicosanoid and eicosanoid-related metabolites with exercise traits[20]. The PC scores for PC1 and PC2 for each exercise trait are displayed in Supplementary Fig. 2. The first two principal components, PC1 and PC2, explained up to 73% of the total variance. We found that

PC1 scores were preferentially greater in magnitude for peak oxygen uptake, chronotropic response, and diastolic and pulmonary vascular reserve, with positive scores for % predicted peak $VO_2$ and % predicted HR, and negative scores for $\Delta PCWP/\Delta CO$ and $\Delta PAP/\Delta CO$. By contrast, PC2 scores were greater for ventilatory efficiency, BP response, and peripheral oxygen extraction with positive scores for $\Delta SBP$ and C(a-v) $O_2$ and negative scores for $VE/VCO_2$ slope.

Of note, we found that PC1 was preferentially loaded on eicosanoids and eicosanoid-related metabolites associated with HFpEF, including positive loadings on metabolites associated with lower odds of HFpEF including 13 HDoHE and 12,13 diHOME and negative loadings on metabolites with higher odds of HFpEF including 12,13 EpOME. By contrast, PC2 was not preferentially loaded on eicosanoid and related metabolites associated with HFpEF.

### Mediation analyses

Of the 70 HFpEF eicosanoids and eicosanoid-related metabolites, we found that 55 were also associated with BMI, 54 with DM, and 53 with HTN ($p < 0.05$ for all) with directionality of associations consistent with potential mediation. We performed formal mediation analyses to examine whether these metabolites might mediate the association of BMI, DM, and HTN with HFpEF. We identified 35 unique eicosanoid and related metabolites (7 named and 28 putative eicosanoids and related metabolites) that mediated the association of BMI with HFpEF, 50 eicosanoid and eicosanoid-related mediators (11 named and 39 putative eicosanoids and related metabolites) of diabetes with HFpEF, and 45 eicosanoid and eicosanoid-related mediators (12 named and 32 putative eicosanoids and related metabolites) of HTN with HFpEF (mediation effect range = 5% to 25%, Table 3, Supplementary Data 6). Notably, among the 7 named eicosanoids and eicosanoid-related mediators, 8(9)-EpETE and 12,13 dihydroxy-9Z-octadecenoic acid (diHOME) explained >10% of the association between BMI, diabetes, and HTN with HFpEF. For example, 8(9)-EpETE and 12,13 diHOME mediated 21% and 18% of the association between BMI and HFpEF, respectively (Fig. 3).

### Association of HFpEF eicosanoids and eicosanoid-related metabolites with incident HF and subtypes in MESA

Among 5192 MESA participants, we observed 283 HF events over a median follow-up time of 17 years (baseline characteristics in Supplementary Data 1). Of the 63 identified HFpEF-related eicosanoids and related metabolites from MGH CPET assayed in the MESA cohort, 18 (3 named and 15 putative eicosanoids or eicosanoid-related metabolites) demonstrated significant associations with incident HF including 11ß-dhk-PGF2α, 11(S)-hydroxy-5Z,8X,12E,14Z,l17Z-eicosapentaenoic acid (HEPE), and 9,10 EpOME (Fig. 4, Supplementary Data 7). Associations with incident HF in the MESA sample were directionally consistent with the primary analysis for 14 of 18 eicosanoids and eicosanoid-related metabolites. Specifically, a 1-SD higher increase in 11ß-dhk-PGF2α concentration was associated with 1.2-fold increased hazard of incident HF (hazard ratio [HR] = 1.24, 95% CI = 1.05–1.47, $p = 0.01$), and a 1-SD higher in 9,10 EpOME concentration was associated with 1.3-fold increased hazards of incident HF (HR = 1.29, 95% CI = 1.10–1.52, $p = 0.002$), while a 1-SD increase in 11(S) HEPE was associated with a 0.8-fold decreased hazards of incident HF (HR = 0.79, 95% CI = 0.69−0.90, $p = 0.0005$). Results were similar after further adjustment for heavy alcohol consumption (Supplementary Data 5).

With respect to HF subtypes, we observed 76 HFrEF events and 67 HFpEF events over a median follow-up time of 17 years. We found that 1 unnamed HFpEF-related eicosanoid metabolite was associated with incident HFrEF and 4 metabolites (1 named, [11(S) HEPE], 3 unnamed) were associated with incident HFpEF (Supplementary Data 8).

### Discussion

We investigated specific eicosanoid and eicosanoid-related metabolites, bioactive lipids that regulate upstream initiation of systemic inflammation, and their associations with HFpEF in a hospital-based sample. Our findings are three-fold: First, we identified 70 distinct eicosanoid and eicosanoid-related metabolites that were associated with HFpEF status, including prostaglandins, epoxides, and oxylipins, with replicated findings for 18 eicosanoids with incident HF in the MESA cohort. Second, eicosanoid and related metabolite profiles characterized distinct exercise manifestations related to HFpEF. Finally, we show that distinct eicosanoids and eicosanoid-related metabolites may mediate the association of cardiometabolic risk factors including BMI, HTN, and DM with HFpEF. Unique aspects of this investigation included the application of a directed nontargeted LC-MS approach to quantify eicosanoid and related metabolites at scale, deployed in a hospital-based sample of individuals with comprehensive deep physiologic phenotyping to study exercise contributions to HFpEF and further corroboration of findings in a large community-based sample. Taken together, our findings highlight important upstream inflammatory pathways associated with HFpEF and HFpEF-related exercise traits that may offer insights into the pathogenesis of HFpEF and potential future therapeutic targets.

Mounting evidence supports the central role of systemic inflammation in the development of HFpEF and its varied manifestations. Specifically, systemic inflammation has been proposed as the common thread linking comorbidities to cardiovascular remodeling and progression to HFpEF. This comorbidity-inflammation paradigm asserts that activation of systemic inflammation, induced by comorbidities including HTN, obesity, and DM, contributes to myocardial inflammation, microvascular and endothelial dysfunction, and pulmonary vascular remodeling, all purported drivers of HFpEF pathogenesis[6]. In prior studies, downstream markers of inflammation including CRP, IL-6, and TNF-α have been associated with incident HFpEF, disease severity, and outcomes[8–10]. However, neither the causal role of inflammatory mediators nor their role as therapeutic targets have been established. The initiation of inflammation in humans is governed by small-molecule derivatives of arachidonic acid and other polysaturated fats (PUFAS), termed eicosanoids[13]. Synthesized by a set of highly conserved enzymes (cyclooxygenases, lipoxygenases, and cytochrome P450 enzymes), eicosanoids and eicosanoid-related metabolites regulate both the activation and suppression of systemic inflammation. Previous studies of eicosanoids and related metabolites in HF demonstrated greater levels of pro-inflammatory metabolites including prostaglandins $PGI_2$ and $PGE_2$, but investigations of have been limited in scale[21,22]. The advent of LC-MS has enabled global profiling of eicosanoids and eicosanoid-related metabolites including products of cytochrome P450 monoxygenases, cyclooxygenases, and lipoxygenases. Leveraging this LC-MS approach, we present the first comprehensive examination of eicosanoid and related metabolites and HFpEF.

Among 890 eicosanoids and eicosanoid-related metabolites assayed, we identified 70 unique eicosanoid and related metabolites associated with HFpEF status, including prostaglandins, oxylipins, eicosatetraenoic acids, docosanoids, resolvins, and other classical and non-classical eicosanoids. Prostaglandin epimers 15R-PGF2α and 11ß-dhk-PGF2α were most strongly associated with HFpEF status. PGF2α, a COX-derived arachidonic acid metabolite, and its receptor, are primarily expressed in the female reproductive system and kidney. In women, PGF2α regulates the development of the corpus luteum via progesterone secretion and stimulation of angiogenic factors, physiologic effects of which have been exploited for pharmaceutical induction of labor and termination of pregnancy. The role of PGF2α in cardiovascular homeostasis is less well characterized, but a previous mouse model found that PGF2α elevated blood pressure and promoted atherogenesis, potentially via modulation of the renin-angiotensin-aldosterone system[23,24]. In this context, the associations of 15R-PGF2α and 11ß-dhk-PGF2α with HFpEF are notable given the established role of HTN in HFpEF pathobiology and female susceptibility to HFpEF, though interestingly the endogenous 15k-PGF2α

**Table 3 | Mediation analyses identified potential eicosanoid and eicosanoid-related metabolite mediators of association of BMI, diabetes, and HTN with HFpEF**

| m/z | RT | Putative ID | Mediation | | Covariate → EIC | | | EIC → HFpEF | | | Direct effect | | | Total effect | | |
|---|---|---|---|---|---|---|---|---|---|---|---|---|---|---|---|---|
| | | | Effect | p-value | ß | SE | p-value | ß | SE | p-value | ß | SE | p-value | ß | SE | p-value |
| **Body mass index** | | | | | | | | | | | | | | | | |
| 377.2305 | 4.8408 | 8(9)-EpETE | 0.21 | <E−16 | −0.35 | 0.04 | 1.41 E−16 | −0.37 | 0.10 | 3.27 E−04 | 0.50 | 0.11 | 3.68 E−06 | 0.63 | 0.10 | 1.22 E−09 |
| 313.2388 | 3.4472 | 12,13 diHOME | 0.18 | <E−16 | −0.31 | 0.04 | 4.94 E−13 | −0.37 | 0.10 | 2.59 E−04 | 0.52 | 0.11 | 9.90 E−07 | 0.63 | 0.10 | 1.22 E−09 |
| 353.2336 | 2.3865 | 11ß-PGE1 | 0.10 | 0.005 | 0.22 | 0.04 | 2.83 E−07 | 0.29 | 0.10 | 0.004 | 0.57 | 0.10 | 7.10 E−08 | 0.63 | 0.10 | 1.22 E−09 |
| **Diabetes** | | | | | | | | | | | | | | | | |
| 377.2305 | 4.8408 | 8(9)-EpETE | 0.25 | <E−16 | −0.57 | 0.12 | 2.03 E−06 | −0.48 | 0.10 | 1.49 E−06 | 0.79 | 0.27 | 0.003 | 1.02 | 0.26 | 9.09 E−05 |
| 313.2388 | 3.4472 | 12,13 diHOME | 0.24 | <E−16 | −0.58 | 0.12 | 1.35 E−06 | −0.46 | 0.10 | 3.68 E−06 | 0.80 | 0.27 | 0.003 | 1.02 | 0.26 | 9.09 E−05 |
| 293.2122 | 3.9713 | 9(S) HOTrE | 0.16 | 0.005 | −0.54 | 012 | 6.15 E−06 | −0.32 | 0.10 | 0.0014 | 0.87 | 0.27 | 0.001 | 1.02 | 0.26 | 9.09 E−05 |
| 293.2123 | 4.1502 | 13(S) HOTrE(y) | 0.15 | 0.002 | −0.50 | 0.12 | 2.80 E−05 | −0.32 | 0.09 | 7.12 E−04 | 0.88 | 0.27 | 9.39 E−04 | 1.02 | 0.26 | 9.09 E−05 |
| 353.2336 | 2.2385 | 11ß-PGE1 | 0.14 | 0.002 | 0.41 | 0.12 | 6.83 E−04 | 0.37 | 0.10 | 3.15 E−04 | 0.90 | 0.27 | 7.12 E−04 | 1.02 | 0.26 | 9.09 E−05 |
| **Hypertension** | | | | | | | | | | | | | | | | |
| 313.2388 | 3.4471 | 12,13 diHOME | 0.20 | <E−16 | −0.50 | 0.09 | 7.42 E−09 | −0.42 | 0.10 | 2.33 E−05 | 0.80 | 0.19 | 2.17 E−05 | 0.98 | 0.18 | 8.98 E−08 |
| 377.2305 | 4.8408 | 8(9)-EpETE | 0.16 | <E−16 | −0.36 | 0.09 | 5.22 E−05 | −0.48 | 0.10 | 2.02 E−06 | 0.86 | 0.19 | 4.94 E−06 | 0.98 | 0.18 | 8.98 E−08 |
| 293.2123 | 4.1502 | 13(S) HOTrE(y) | 0.12 | 0.004 | −0.45 | 0.09 | 2.78 E−07 | −0.29 | 0.10 | 0.003 | 0.86 | 0.18 | 3.73 E−06 | 0.98 | 0.18 | 8.98 E−08 |
| 293.2122 | 3.9713 | 9(S) HOTrE | 0.12 | 0.012 | −0.43 | 0.09 | 1.14 E−06 | −0.30 | 0.10 | 0.004 | 0.87 | 0.19 | 3.25 E−06 | 0.98 | 0.18 | 8.98 E−08 |
| 353.2336 | 2.3865 | 11ß-PGE1 | 0.10 | 0.003 | 0.30 | 0.09 | 6.15 E−04 | 0.36 | 0.10 | 5.42 E−04 | 0.90 | 0.19 | 1.42 E−06 | 0.98 | 0.18 | 8.98 E−08 |

Metabolites displayed were found to mediate >10% of the association between body mass index, diabetes, or hypertension with HFpEF.

Mediation effect represents the proportion of the effect of covariate on HFpEF that is mediated by the specific eicosanoid.

Direct effect represents the residual effect of the covariate on HFpEF independent of mediated effects from the specific eicosanoid. Total effect represents the total effect of the covariate on HFpEF. p < 0.05 considered significant mediation effect based on mediation analyses. Standardized ß-coefficients are shown.

CI confidence interval, DiHDPA dihydroxy-4Z,7Z,10Z,13Z,16Z-docosapentaenoic acid, diHOME dihydroxy-9Z-octadecenoic acid, EIC eicosanoid, EpETE epoxy-5Z,11Z,14Z,17Z-eicosatetraenoic acid, EpETE epoxy-5Z,11Z,14Z,17Z-eicosatetraenoic acid, EpOME epoxy-9Z-octadecenoic acid, HFpEF heart failure with preserved ejection fraction, m/z mass to charge ratio, PGF2α prostaglandin F2α, RT retention time, SE standard err.

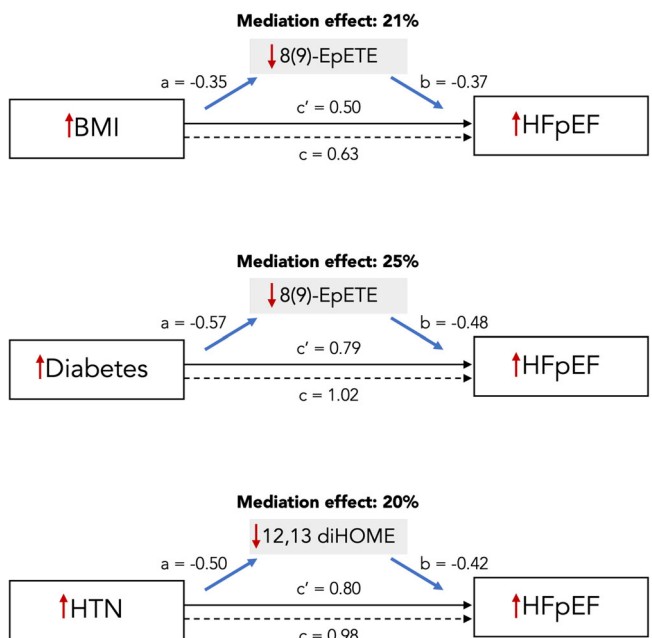

**Fig. 3 | Eicosanoid mediation model for the association of body mass index, diabetes, and hypertension with HFpEF.** Mediation analyses identified 3 eicosanoid metabolites with known molecular identity, which were found to mediate >10% of the association of body mass index, diabetes, and hypertension with HFpEF, including 8(9)-EpETE and 12,13 diHOME. We display the eicosanoid mediation model for the most significant eicosanoid mediator of the association of body mass index, diabetes, and hypertension with HFpEF. In this figure, *a* represents the effect of the covariate on eicosanoid, *b* represents the effect of eicosanoid on HFpEF, *c* represents the total effect of the covariate on HFpEF, and *c′* is the residual direct effect of the covariate on HFpEF (independent of mediated effects from eicosanoid). Red arrows represent direction of effect of association (e.g. BMI is positively associated with the eicosanoid mediator and with HFpEF. The eicosanoid mediators are negatively associated with HFpEF). Mediation effect considered significant if mediation effect *p*-value threshold < 0.05 by mediation analysis. Standardized beta-coefficients are shown. BMI body mass index, HFpEF heart failure with preserved ejection fraction.

metabolite was associated with lower odds of HFpEF[25–27]. Little is known about the distinct biological activities of endogenous PGF2α vs its epimers particularly in the context of HFpEF risk, but the established role of another PGFα epimer 8-iso-PGF2α as a marker of oxidative stress highlights the unique biological properties of PGF2α epimers that may explain the discordant associations observed between endogenous PGF2α and PGF2α epimers with HFpEF status[28,29]. By contrast, epoxide 8(9)-EpETE demonstrated the greatest negative association with HFpEF. Epoxy-eicosatetraneoic acids (EpETE) are cytochrome P450-derived products of the cardioprotective ω-PUFAs eicosapentaneoic acid (EPA) and docosahexaenoic acid (DHA) that induce vasodilation, stimulate angiogenesis, and suppress inflammation. The beneficial effects of ω−3 PUFAs have been demonstrated in both observational studies and clinical trials, with notable reductions in hospital admission and death among patients with chronic HF treated with ω−3 PUFAs vs placebo[30,31]. We also identified resolvin D1 as a key eicosanoid metabolite associated with lower odds of HFpEF. Resolvins are important EPA and DHA derivatives that promote the resolution of inflammation. Resolvin D1 specifically has been shown to exert resolution of post-myocardial infarction inflammatory injury and delayed the onset of ventricular dysfunction and heart failure in a mouse model of myocardial infarction[32]. The association of resolvin D1 with lower odds of HFpEF provides further support for the inflammatory basis of HFpEF pathogenesis and highlights resolvin D1 as a potential target for future HF intervention.

Recognizing that HFpEF is a highly heterogeneous condition characterized by the hallmark feature of exercise intolerance, we further dissect the association of eicosanoid profiles with cardiac and extra-cardiac contributors to exercise intolerance. Identifying biologic pathways underlying exercise deficits may enable development of therapies for HFpEF more specifically targeted at subphenotypes[33]. In this context, we specifically examined cardiac and extracardiac exercise responses and found that exercise traits carried distinct eicosanoid profiles that collectively contributed to the metabolite profile of overall HFpEF status. For example, 12,13 EpOME and 11ß-dhk PGF2α, two HFpEF-related eicosanoids, were uniquely associated with PCWP/CO, a marker of diastolic reserve. Similarly, 8(9)-EpETE was associated with % predicted peak VO2 and % predicted HR, but not with other queried exercise traits. Most data examining eicosanoids with exercise traits have focused on peak VO2, with early studies demonstrating significant associations of prostaglandin PGI2 and its hydration product 6-keto-PGF1α with VO2 max[34,35]. We too found significant associations of prostaglandins and prostaglandin-like metabolites with peak VO2, but in contrast to the early studies that demonstrated positive associations with VO2 max, prostaglandin metabolites including 15-R PGF2α and 6ß-PGI1 were negatively associated with % predicted peak VO2 in our sample in keeping with the hypothesis that inflammation promotes development of exercise intolerance in HFpEF. However, prostaglandins, while long regarded as pro-inflammatory metabolites, are known to exert additional anti-inflammatory and vasodilatory effects that are context dependent and may explain the discordant findings[36]. More recently, a lipidomic analysis found that moderate intensity exercise dramatically increased circulating levels of oxylipin 12,13-diHOME in both mouse and human subjects[37]. Investigators further demonstrated that surgical removal of brown adipose tissue in mice reversed the exercise-induced increase in 12,13 diHOME levels and treatment with 12,13-diHOME increased skeletal muscle fatty acid uptake and oxidation. In our sample, we showed that 12,13-diHOME was strongly associated with both % predicted peak VO2 and % predicted HR, pointing to the important role of 12,13-diHOME and related pathways in the development of exercise intolerance in HFpEF.

While our investigation is the first large-scale evaluation of eicosanoids and eicosanoid-related metabolites in HFpEF, eicosanoid and related metabolites have been previously associated with clinical precursors of HFpEF including obesity, HTN, and diabetes[38–40]. For example, in early experimental studies, investigators observed higher baseline levels of urinary 11-dehydro-thromboxane B2 (11dgTxB2), an eicosanoid metabolite and in vivo index of platelet activation, among patients with type II diabetes vs controls[41]. Notably, metabolic control with insulin therapy and low-dose aspirin reduced levels of 11dgTxB2 by nearly 50% and 80%, respectively. Similarly, levels of F(2)-isoprostane, a class of eicosanoids involved in lipid peroxidation and oxidative stress, were increased in patients with diabetes and significantly reduced with improved metabolic control and vitamin E supplementation[38]. LC-MS profiling has extended these initial findings and enabled identification of metabolites associated with HTN and DM. In a study of 8099 healthy participants from the FINRISK 2002 cohort and 2859 participants from the Framingham Heart Study, LC-MS profiling of 545 eicosanoid metabolites and related oxylipin mediators identified 187 unique metabolites with SBP[18]. An eicosanoid risk score comprised of 6 metabolites independently associated with SBP, derived from the FINRISK 2002 discovery cohort, accurately discriminated odds of HTN with >2-fold increased odds of HTN among individuals with highest vs lowest quartile eicosanoid risk. A complementary evaluation of eicosanoids and related metabolites and DM in the FINRISK 2002 cohort found 76 eicosanoids and eicosanoid-related metabolites individually associated with incident type 2 diabetes[42]. A three-eicosanoid risk score, comprised of 8-iso-prostaglandin A1 (8-iso-PGA1), 12-hydroxy-5,8,10-heptadecatrienoic

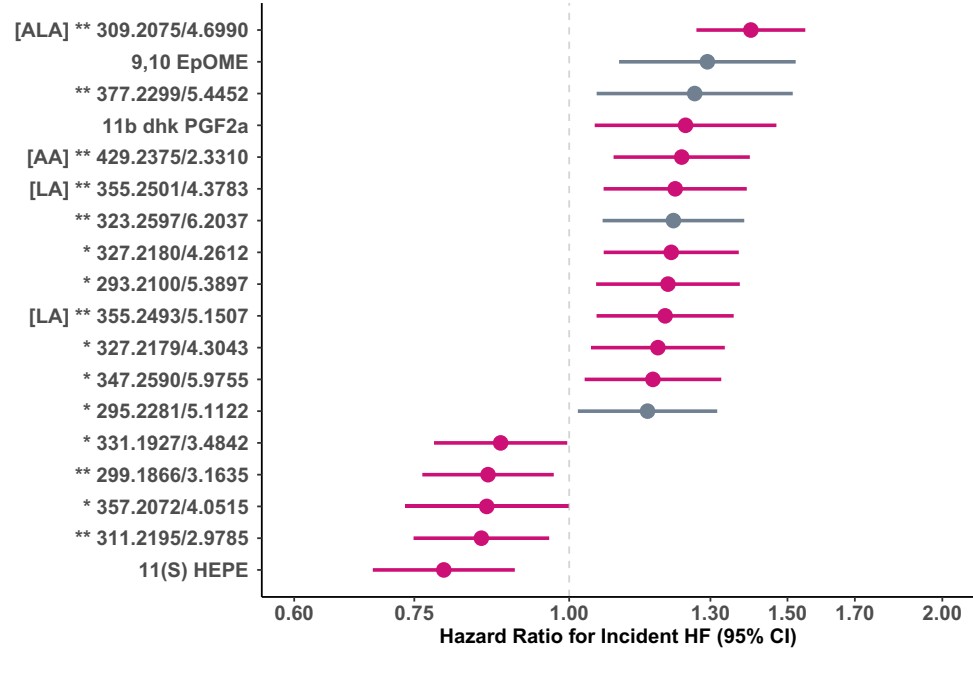

**Fig. 4 | Association of HFpEF-related eicosanoids and eicosanoid-related metabolites (from MGH CPET) with incident HF in MESA.** Eicosanoids and eicosanoid-related metabolites chosen for analysis in MESA were significantly associated with HFpEF status in the MGH CPET cohort. Hazards ratios displayed represent hazards of developing HF per 1-SD increase in log-transformed eicosanoid and related metabolite in MESA. Data presented are HR (dots) with 95% confidence intervals (bars). Analyses are adjusted for age, sex, site, and plate number. $p$-value threshold < 0.05 considered significant based on Cox regression models. Eicosanoids and eicosanoid-related metabolites with directionally consistent associations in both the MGH CPET and MESA samples are denoted in red. Eicosanoids and eicosanoid-related metabolites with directionally discordant associations in the MGH CPET and MESA samples are denoted in gray. *denotes putative eicosanoid metabolite. **denotes putative eicosanoid-related metabolite. CI confidence interval, HF heart failure, ALA alpha-linoleic acid, LA linoleic acid. Source data are provided as a Source Data file.

acid (12-HHTrE), and an unidentified eicosanoid, was associated with a >50% increased hazards of incident diabetes.

Given the purported inflammatory link between cardiometabolic disease and HFpEF, we examined whether eicosanoids and eicosanoid-related metabolites potentially mediate the association between cardiometabolic risk factors (BMI, HTN, and DM) with HFpEF. In mediation analyses, we identified 6 eicosanoid and related metabolites as potential mediators, including prostaglandins, oxylipins, epoxides, docosanoids, and ω−3 PUFAs. Two metabolites specifically, oxylipin 12,13 diHOME and epoxide 8(9)-EpETE, were found to negatively mediate >10% of the association between BMI, DM, and HTN with HFpEF. Specifically, in mediation analyses, we showed that greater BMI and presence of DM and HTN were associated with greater odds of HFpEF via lower levels of 12,13 diHOME and 8(9)-EpETE. As previously discussed, increased 12,13 diHOME levels, induced by physical exercise or other stimuli like exposure to cold, has been associated with improved metabolic health. Its identification as a potential mediator of the relationship between cardiometabolic risk factors and HFpEF highlights the intimate relationship between inflammation, obesity, and metabolic changes related to physical exercise with HFpEF development[43]. Moreover, strategies to increase levels of 12,13-diHOME hold great promise for the prevention and treatment of HFpEF. Similarly, the discovery of 8(9)-EpETE as a mediator of cardiometabolic comorbidities with HFpEF highlights ω−3 PUFAs as a potential target for therapeutic intervention in HFpEF.

Our study has several limitations. First, the variance of measured eicosanoid and eicosanoid-related metabolites is known to be significantly influenced by sample processing. Our samples were stored and processed using an established protocol, and all analyses were performed using standardized rather than absolute values of

metabolites[44]. Second, we studied patients referred for clinically indicated level 3 CPET. While this allowed us to define HFpEF using rigorous physiologic criteria with careful assessment of exercise physiology, we acknowledge that referral bias may limit generalizability to other HFpEF samples, including HFpEF registries or clinical trials. In that context, we further note individuals in MGH CPET who did not meet physiologic HFpEF criteria are not necessarily 'healthy' controls. Third, there were substantial differences in baseline characteristics between participants with and without HFpEF that may have contributed to observed associations between eicosanoid metabolites and HFpEF. To account for differences in comorbidity profiles between HFpEF and control subjects, we matched participants by key covariates age, BMI, and HTN, and found that findings were consistent with the primary analyses. Fourth, the MGH CPET and MESA incident HF cohorts represented different patient populations, and necessitated alignment of eicosanoid metabolites across different samples. The different patient characteristics may explain the directionally discordant associations observed between several eicosanoid metabolites and HF in the MGH CPET and MESA analyses. For example, 9,10 EpOME was associated with lower odds of HFpEF within the MGH CPET cohort but strongly associated with incident HF in MESA. Moreover, we acknowledge that our biological understanding of eicosanoids as both potential substrates and products of interlinked enzymatic pathways is yet to be characterized, and that our analyses may not capture complex relationships along biological pathways. Fifth, given limited number of HF events in the MESA sample, we were not powered to specifically examine incident HFpEF or to perform mediation analyses to examine whether eicosanoids may mediate the association of clinical risk factors with incident HFpEF. Finally, accurate identification and classification of eicosanoid and related metabolites using the LC/

MS platform is inherently challenging, but MS signals have been consistently mapped to known and putative eicosanoids in previous investigations. We discovered a number of novel eicosanoid and related metabolites associated with HFpEF and acknowledge that exact molecular identity is not known for all of our findings at this time, and that future studies will be needed to identify exact identities.

Among patients with chronic dyspnea and preserved LVEF, 70 pro- and anti-inflammatory eicosanoid and eicosanoid-related metabolites were associated with HFpEF status. Specifically, prostaglandin and linoleic acid derivatives portended greater odds of HFpEF, whereas specific epoxides, oxylipins, DHA and prostaglandin derivatives and resolvins were associated with lower odds of HFpEF. Importantly, among these HFpEF-related eicosanoid and related metabolites, 18 were also associated with incident HF in the community. Distinct eicosanoid profiles further characterized discrete contributors to exercise intolerance in HFpEF and were found to act as potential mediators in the association of cardiometabolic traits and HFpEF. Further interrogation of HFpEF-related eicosanoids and associated pathways may provide insights into HFpEF pathophysiology and offer potential targets for therapeutic intervention.

## Methods

### Ethics statement
All participants provided written informed consent. The MGH CPET analysis was approved by the Massachusetts General Brigham Institutional Review Board and the MESA analysis was approved by institutional review boards of each of the participating field sites (Johns Hopkins University, Baltimore, MD; Northwestern University, Chicago, IL; Wake Forest University, Winston-Salem NC; University of California, Los Angeles, CA; Columbia University, New York City, NY; University of Minnesota, Minneapolis, MN).

### Study cohorts
The Massachusetts General Hospital (MGH) CPET study included individuals with chronic dyspnea (New York Heart Association Class II-IV symptoms) and preserved LVEF (LVEF ≥ 50%) who underwent clinically indicated cardiopulmonary exercise testing (CPET) with invasive hemodynamic monitoring at the MGH between 2009 and 2017. Plasma sampling was performed on $n = 644$ participants. From this sample, we excluded participants with history of cardiac or pulmonary transplant ($n = 15$), complex congenital heart disease ($n = 15$), mitochondrial disorder ($n = 12$), pulmonary arterial hypertension ($n = 10$), undergoing evaluation for lung transplant ($n = 10$), significant valvular disease ($n = 51$), and severe lung disease ($n = 21$), yielding a final sample of $n = 510$ included in the MGH CPET cohort.

The Multi-Ethnic Study of Atherosclerosis (MESA) is a population-based longitudinal cohort study of White, Black, Hispanic, and Chinese individuals recruited from six U.S. communities (Baltimore City and Baltimore County, MD; Chicago, IL; Forsyth County, NC; Los Angeles County, CA; Northern Manhattan and the Bronx, New York City, NY; and St. Paul, MN). For this study, we included all MESA participants who participated in examination cycle 2 (2002-2004) with available eicosanoid metabolite assays ($n = 5457$). We excluded individuals with prevalent HF ($n = 25$), advanced chronic kidney disease ($n = 17$), and those with missing covariates ($n = 223$), yielding a final sample of $n = 5192$ in the MESA cohort.

### Clinical assessment and biomarkers
Participants in the MGH CPET Cohort underwent medical history and physical examination (including measurement of vital signs and body mass index), and fasting blood draw at the time of CPET. All MESA participants underwent comprehensive medical history, physical examination, laboratory testing, and anthropometric measurements at each examination. Sex was self-identified and determined based on clinical chart review. We considered sex as a clinical covariate in all multivariable models. To ensure that primary analyses are sufficiently powered in a modest sample, sex-pooled analyses were performed rather than stratifying by sex. Body mass index (BMI) was defined as weight (kg)/height$^2$ (m$^2$) and obesity was defined as BMI ≥ 30 kg/m$^2$. Advanced chronic kidney disease was defined as an estimated glomerular filtration rate <30 mL/min/1.73m$^2$. Smoking was defined as self-reported daily use of cigarettes and former smoking as a history of smoking. We defined HTN as a physician diagnosis of HTN or use of antihypertensive medications and DM as a physician diagnosis of DM or use of diabetic medications. Biomarkers N-terminal pro-B-type natriuretic peptide (NT-pro BNP) and high-sensitivity C reactive protein (hsCRP) were measured on fasting blood samples. NT-pro BNP was ascertained using an electrochemiluminescence immunoassay (Roche, intra-assay coefficient of variation [CV] = 2.4%–3.8%) and hsCRP was measured via an immunoturbidimetric assay (Roche, CV = 0.4%–8.4%).

### Cardiopulmonary exercise testing
In MGH CPET, all participants underwent pulmonary artery catheter insertion via the internal jugular vein followed by maximal upright cycle ergometry with a previously described ramp protocol (3-min initial period of unloaded exercise followed by a 5–20 watt/minute continuous ramp). Serial gas exchange (MedGraphics, St. Paul, MN) and minute-by-minute hemodynamic measures were obtained at rest and during each minute of exercise. Gas exchange and hemodynamic measures of interest included peak $VO_2$, % predicted $VO_2$ as ascertained via the Wasserman equation[45], continuous respiratory exchange ratio (RER), VE/VCO$_2$ slope, systolic and diastolic blood pressures, pulmonary arterial pressure (PAP), pulmonary capillary wedge pressure (PCWP), arterial-venous $O_2$ difference (C[a-v]$O_2$), and direct Fick cardiac output (CO) values. Diastolic response to exercise was ascertained from serial PCWP and CO measurements to calculate ΔPCWP/ΔCO slope, with a value > 2.0 mmHg/L/min considered abnormal[46]. Pulmonary vascular response to exercise was calculated as ΔPAP/ΔCO slope and defined as abnormal if ΔPAP/ΔCO > 3 mmHg/L/min[47].

### Definition of HFpEF
In MGH CPET, HFpEF was defined using physiologic criteria including (1) evidence of elevated left ventricular filling pressures at rest (resting supine PCWP ≥ 15 mmHg) or during exercise (peak exPCWP ≥ 15 mmHg and abnormally steep increase in the PCWP relative to CO during exercise (ΔPCWP/ΔCO slope >2.0 mmHg/L/min) and (2) impaired peak oxygen consumption (% predicted peak $VO_2$ < 80%).

### Clinical assessment in MESA
In MESA, BMI was defined as weight (kg)/height$^2$ (m$^2$) and obesity was defined as BMI ≥ 30 kg/m$^2$. End-stage renal disease was defined as an estimated glomerular filtration rate <30 mL/min/1.73m$^2$. Smoking was defined as self-reported daily use of cigarettes and former smoking as a history of smoking. HTN was defined as a SBP of ≥140 mmHg, a DBP of ≥90 mmHg, or self-reported use of antihypertensive medications. DM was defined as a fasting blood glucose ≥ 126 mg/dL and/or self-reported history of a physician-diagnosis of DM, or the use of diabetic medications.

### Ascertainment of HF outcomes in MESA
In MESA, adjudication of HF events was performed using previously detailed protocols by study investigators after review of self-reported data (obtained via telephone or in-person interview every 9–12 months) and hospital admissions. We considered probable or definite hospitalized HF events. Probable HF was defined as a diagnosis of HF by a physician and medical treatment of HF. A designation of definite HF further required additional objective evidence of HF including pulmonary edema on chest radiography, reduced left ventricular function by echocardiography or ventriculography, or evidence of LV diastolic dysfunction.

## Profiling of eicosanoids and eicosanoid-related metabolites

Fasting blood samples were drawn after a minimum of 8 h of fasting, and subsequently processed and stored at −80 °C. Eicosanoid profiling of plasma samples for both the discovery and validation cohorts was performed at the University of California, San Diego. Briefly, we identified eicosanoid and eicosanoid-related metabolites from participant plasma samples using a directed, non-targeted LC-MS approach combined with computational chemical networking of spectral fragmentation patterns, as described previously[17,44]. Following extraction of eicosanoid and eicosanoid-related metabolites from plasma samples, 20 mL of prepared sample was injected onto a Phenomenex Kinetex C18 (1.7 μm particle size, 2.1 × 100 mm) column that was heated to 50 °C. Chromatographic separation was achieved using a Thermo Vanquish UPLC by ramping mobile phase A (70% water, 30% acetonitrile and 0.1% acetic acid) to mobile phase B (50% acetonitrile, 50% isopropanol, 0.02% acetic acid) using the following gradient profile: 1% B from 0 to 0.25 min, 1% to 55% B from 0.25 to 5 min, 55% to 99% B from 5 to 5.50 min, and 99% B from 5.50 to 7.5 min. A one-minute re-equilibration at 1% B was performed after each injection. Parameters set included: flow rate of 0.375 mL/min, needle wash 5 s post-draw (50:25:25:0.1 water:acetonitrile:isopropanol:acetic acid), and seal wash set to periodic (50:50 water:isopropanol). Detection was performed on a coupled Thermo QExactive orbitrap mass spectrometer equipped with a heated electrospray ionization (HESI) source and collision-induced dissociation fragmentation (CID). Source geometry was optimized for signal stability and intensity about the $x$, $y$, and $z$ axis by infusing CUDA and PGA2 standards into the mobile phase stream at 1%, 50% and 99% B with a final probe height of 0.75 B-C, x-offset of 1 mm left of center, and a Y-offset of 1.70 according to the source micrometer. The source settings applied to the samples were: negative ion mode profile data, sheath gas flow of 40 units, aux gas flow of 15 units, sweep gas flow of 2 units, spray voltage of −3.5 kV, capillary temperature of 265 °C, auxiliary gas temperature of 350 °C, and S-lens RF of 45. An MS1 scan event was followed by 4 DDA scan events using an isolation window of 1.0 $m/z$ and a normalized collision energy of 35 arbitrary units with active exclusion set to 5 s. A scan range of $m/z$ 225–650, mass resolution of 17.5k, AGC of 1e6, and inject time of 50 ms was used for MS1 scan events and a mass resolution of 17.5 K, AGC 3e5, and inject time of 40 ms was used for tandem MS acquisition. Finally, additional tandem MS spectra were collected in a targeted fashion using inclusion lists to obtain high quality MS2 spectra for each compound once prioritized oxylipins were discovered. Data for specific eicosanoid and related metabolites including lipid standards, MS characteristics, and naming are outlined in the LIPID MAPS database. MSconvert version 3.0.9393 was used for file conversion from RAW to mzXML. In-house R scripts were used to perform initial bulk feature alignment[48]. mzMine 2.21 was used to perform feature extraction, secondary, alignment, and compound identification. R (version 3.3.3) was used to perform all statistical analyses related to LC/MS methods.

Data for specific eicosanoid and related metabolites including lipid standards, MS characteristics, and naming are outlined in LIPID MAPS[49]. From MGH CPET cohort, 936 eicosanoid and eicosanoid-related metabolites were first identified. Eicosanoid metabolites with >90% missing data were excluded from analysis. For the remaining metabolites (n = 890), missing values below the detection threshold were imputed as 25% of the minimum value detected for each eicosanoid and eicosanoid-related metabolite. In the MESA analyses, eicosanoids and related metabolites were then matched between the MGH CPET and MESA samples by comparing LC-MS profiles. Of the 70 HFpEF-related eicosanoid and eicosanoid-related metabolites identified in the MGH CPET sample, 62 were aligned to metabolites in MESA and used for incident HF analyses. Specific metabolites were validated using spectral fragmentation pattern networking and manual annotation. The mean coefficient of variation for radiolabeled internal standard metabolites was 10.2% with a range of 8.1%–13.1%.

## Statistical analyses

Baseline characteristics were summarized for both the MGH CPET and MESA samples. Results are reported as means (standard deviation) or medians (inter-quartile ranges) for continuous variables and frequencies (percentages) for categorical variables. Eicosanoid and related metabolite concentrations were natural log-transformed due to right-skewed distributions and then standardized to a mean of 0 and standard deviation of 1. In primary analyses, we examined the cross-sectional association between single eicosanoid metabolites with HFpEF status in MGH CPET using logistic regression analyses (Fig. 5). Among eicosanoid and eicosanoid-related metabolites that were significantly associated with HFpEF status in the primary analyses, we then examined their associations with exercise traits using linear regression analyses in secondary analyses. Exercise traits of interest included % predicted peak VO₂, BP response (ΔSBP), chronotropic response (% predicted heart rate), pulmonary vascular reserve (ΔPAP/ΔCO), diastolic reserve (ΔPCWP/ΔCO), ventilatory efficiency (VE/VCO₂), and peripheral oxygen extraction (C[a-v]O₂). All models were adjusted for age, sex, and plate number in primary analyses. Secondary models further adjusted for BMI, diabetes, HTN, present smoking, prevalent myocardial infarction, statin use, aspirin use, and diuretic use. Exploratory models further adjusted for estimated glomerular filtration rate and alcohol use (for MESA analyses only). To account for multiple testing in single eicosanoid models, a false discovery rate (FDR) $q$-value < 0.10 was deemed significant and $p$-value < 0.05 was considered suggestive.

To better understand the associations of potentially correlated eicosanoid and related metabolites with exercise traits, we used principal component analysis (PCA) to examine the association of 890 inter-correlated eicosanoid and eicosanoid-related metabolites with continuous exercise traits. We examined the associations of each of the exercise traits with the first two PC terms (PC1 and PC2) that explained the greatest proportion of the total variance.

Because systemic inflammation is thought to causally mediate the association of cardiometabolic risk factors with HFpEF, we sought to examine whether eicosanoid and related metabolites might mediate the association of specific risk factors with HFpEF status. In exploratory analyses, we conducted mediation analyses by modeling cardiometabolic risk factors (BMI, DM, HTN) as the primary exposure, HFpEF status as the outcome, with eicosanoids and eicosanoid-related metabolites as potential mediators. We performed mediation analyses using quasi-Bayesian approximation (mediate function in the mediation library in R, version 4.5.0)[50,51]. Mediation analyses were performed for eicosanoids that met the following condition: $(x\_m\_beta * m\_y\_beta * x\_y\_beta) \geq 0$. Statistical analyses were performed using the mediate function in the mediation package in R that employed quasi-Bayesian approximation to estimate the indirect, direct, and total effects. We set 5000 Monte Carlo draws for the quasi-Bayesian approximation.

Finally, to further explore the clinical relevance of eicosanoid and related metabolites identified in primary analyses, we took eicosanoid and related metabolites associated with HFpEF status in MGH CPET and examined their association with incident HF and HF subtypes using Cox proportional hazards models in the MESA sample accounting for competing risks of death, other HF subtype, and unclassified HF. All models were adjusted for age, sex, site, and plate number. For the incident HF analyses, a $p$-value < 0.05 was considered significant.

All statistical analyses were performed using R version 4.1.1. We used R package "heatmaply" version 1.3.0 to generate the dendrogram and heatmap. We used the default setting for the heatmaply function which results in distant method to be "Euclidean".

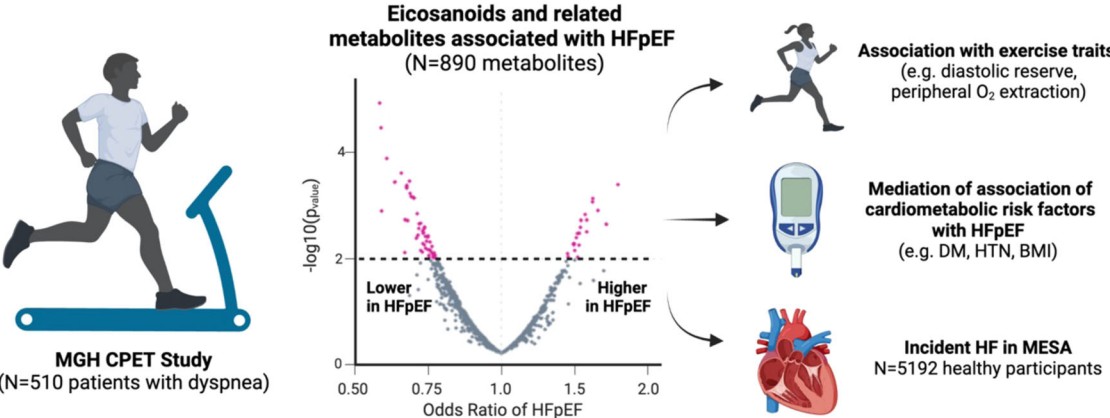

**Fig. 5 | Eicosanoid and eicosanoid-related inflammatory mediators are associated with HFpEF and related exercise traits.** A directed nontargeted LC-MS lipidomics approach identified 70 distinct eicosanoid and eicosanoid-related metabolites that were associated with HFpEF status, of which 18 were also associated with the development of HF in the MESA community cohort. Further examination of the 70 HFpEF-related eicosanoid and related metabolites showed associations with distinct exercise traits characteristic of HFpEF and potential mediation of the association of cardiometabolic risk factors with HFpEF. Created with BioRender.com.

## Reporting summary

Further information on research design is available in the Nature Portfolio Reporting Summary linked to this article.

## Data availability

All raw LC-MS mass spectrometry data files have been deposited in the public UCSD massive data repository (https://massive.ucsd.edu/) under dataset identifier MSV000092775 and can be directly accessed at https://massive.ucsd.edu/ProteoSAFe/dataset.jsp?task=e17abc86a3254eb09d0147b4473369fc. All MESA data including clinical data and individual eicosanoid and related metabolite data can be obtained by application to dbGAP with accession number phs000209.v2.p1 or can be obtained via formal application to the MESA Publications & Presentations (P&P) Committee (https://www.mesa-nhlbi.org/). The MESA P&P committee will review the application as per usual procedures. The generated MGH CPET clinical data are considered sensitive patient data and can therefore not be publicly available in compliance with HIPAA and according to limitations included in the informed consents signed by the study participants. Restricted access for the CPET data can be obtained via the MGH Institutional Review Board (protocol 2017P001587). Requests will be forwarded to the lead author Dr. Emily Lau (elau6@mgh.harvard.edu). Requests should include name and contact details of the person requesting the data, which data and clinical variables are requested and the purpose of requesting the data. Requests will be subject to consideration by the PI of the cohort, Dr. Gregory Lewis, and the members of the MGH Institutional Review Board. Time frame for a response will be within 6 months. Data requests under agreement will be considered for purposes of reproducing the data and subject to appropriate confidentiality obligations and restrictions. The source data (including data used for the creation of Figs. 1, 2, and 4) are provided with this paper. Source data are provided with this paper.

## Code availability

Data processing scripts used to perform the analyses described are available at: DOI: 10.5281/zenodo/8206398 https://github.com/athar71/Eicosanoids_HFpEF.

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

## Acknowledgements

E.S.L. is supported by grants from the National Institutes of Health K23-HL159243 and the American Heart Association #853922. R.V.S. is supported by grants from the National Institutes of Health and the American Heart Association. S.C. is supported by grants from the National Institutes of Health R01-HL143227, R01-HL151828, R01-HL131532, U54-AG065141. J.E.H. is supported by grants from the National Institutes of Health R01-HL134893, R01-HL140224, R01-HL160003, and K24-HL153669. MESA is conducted and supported by the National Heart, Lung, and Blood Institute (NHLBI) in collaboration with MESA investigators. Support for MESA is provided by contracts N01-HC95159, N01-HC-95160, N01-HC-95161, N01-HC-95162, N01-HC-95163, N01-HC-95164, N01-HC-95165, N01-HC95166, N01-HC-95167, N01-HC-95168, N01-HC-95169 and CTSA UL1-RR-024156.

## Author contributions

E.S.L., A.R. and J.E.H. designed the study. E.S.L., A.R., S.Z. and D.W. carried out the analyses. M.A., S.C. and G.D.L. provided critical feedback and framing. All authors (J.S.G., N.A., V.V., M.N., R.V.S., J.A.C.L., S.J.S., B.Y. and M.J.) contributed to writing and editing of the manuscript.

## Competing interests

E.S.L. previously served on the advisory board for Astellas Pharma, unrelated to this work. S.Z.'s contributions to the manuscript reflect his work as a postdoctoral fellow at the Massachusetts General Hospital. The opinions and work presented do not represent Google LLC. R.V.S. is a co-inventor on a patent for ex-RNA signatures of cardiac remodeling with Cytokinetics, which is unrelated to this work. J.E.H. has received research funding from Bayer, AG unrelated to this work. M.J. currently holds equity and a leadership position at Sapient Bioanalytics, LLC, and is engaged in research related to the current study. The remaining authors declare no competing interests.

## Additional information

[1]Division of Cardiology, Department of Medicine, Massachusetts General Hospital, 55 Fruit Street, Boston, Massachusetts 02114, USA. [2]Cardiovascular Research Center, Massachusetts General Hospital, 55 Fruit Street, Boston, MA 02114, USA. [3]CardioVascular Institute, Division of Cardiology, Department of Medicine, 330 Brookline Avenue, Beth Israel Deaconess Medical Center, Boston, MA 02215, USA. [4]Google LLC, 1600 Amphitheatre Parkway, Mountain View, CA 94043, USA. [5]Department of Biostatistics, Boston University School of Public Health, 715 Albany Street, Boston, MA 02118, USA. [6]Department of Preventive Medicine, Northwestern University Feinberg School of Medicine, 420 East Superior Street, Chicago, IL 60611, USA. [7]Division of Cardiology, Department of Medicine Johns Hopkins University School of Medicine, 733 North Broadway, Baltimore, MD 21205, USA. [8]Cardiology Division, Boston University School of Medicine, 715 Albany Street, Boston, MA 02118, USA. [9]Vanderbilt Clinical and Translational Research Center (VTRACC), Vanderbilt University Medical Center, 1211 Medical Center Drive, Nashville, TN 37232, USA. [10]Division of Cardiology, Department of Medicine, Northwestern University Feinberg School of Medicine, 420 East Superior Street, Chicago, IL 60611, USA. [11]Feinberg Cardiovascular Research Institute, Northwestern University Feinberg School of Medicine, 420 East Superior Street, Chicago, IL 60611, USA. [12]Department of Epidemiology, Human Genetics and Environmental Sciences, University of Texas Health School of Public Health, 1200 Pressler Street, Houston, TX 77030, USA. [13]Division of Pulmonary and Critical Care and Sleep Medicine, University of California San Diego, 9500 Gilman Drive, La Jolla, CA 92093, USA. [14]Department of Cardiology, Smidt Heart Institute, Cedars-Sinai Medical Center, 127 South San Vincente Pavilion, Los Angeles, CA 90048, USA. [15]Department of Medicine and Department of Pharmacology, University of California San Diego, 9500 Gilman Drive, La Jolla, CA 92093, USA. [16]These authors contributed equally: Emily S. Lau, Athar Roshandelpoor. ✉e-mail: jho@bidmc.harvard.edu

