## [Peer Review File · Nature Communications]

REVIEWER COMMENTS

Reviewer #1 (Remarks to the Author):

In the current manuscript Lau et al assessed differences in fatty acid-derived eicosanoid metabolites and eicosanoid-related metabolites in HFpEF patients and in HFpEF patients following exercise. The paper reports identifying 70 pro- and anti-inflammatory eicosanoid and eicosanoid-related metabolites associated with HFpEF status. As well, distinct eicosanoid profiles characterized contributors to exercise intolerance and potentially act as mediators associated with cardiometabolic traits and HFpEF.

- The authors have assayed a large amount of metabolite data from MGH patients and MESA cohort study, as well from individuals following exercise. As this is the first study assessing eicosanoid and eicosanoid-related metabolites in HFpEF patients it is noteworthy and a strength of the paper.

- However, many of the metabolites identified were unnamed and considered 'putative' dampening the enthusiasm. Will all the data be deposited in the supplemental files?

- The authors discuss linking comorbidities to CV remodeling and progression of HFpEF in regards to systemic inflammation. However, as the authors did not assess markers of inflammation directly. Is this feasible? As well, what was the impact of exercise on the immune response?

- Can the authors obtain information such as the medication profile of the patients? This might provide information regarding potential effects on key enzymes involved in the metabolism producing eicosanoids (COX, LOX, CYP)

- The correlational findings with 8(9)-EpETE and 12,13-diHOME is interesting and provides novel insight into potential ideas for mechanisms

Reviewer #3 (Remarks to the Author):

The original article by Lau and colleagues investigates the role of eicosanoid and eicosanoid-related metabolites in relation to HFpEF. In total, 70 pro- and anti-inflammatory eicosanoid and eicosanoid-related metabolites were found to be associated with HFpEF. This is an elegantly performed study,

which could serve as a basis for therapeutic interventions for HFpEF. However, multiple points need to be addressed:

- In the baseline characteristics of both studies, MGH CPET and MESA no data on kidney function is provided. Although the authors excluded patients with renal function below 30 ml/min/1.73m² in the MESA study, data on renal function, including creatinine, eGFR, and proteinuria should be provided of both studies. With various different forms of CKD, it is known that many eicosanoids are upregulated, such as PGE₂. Also some specific forms of CKD, such as ADPKD have strongly upregulated eicosanoids. Inflammation is present even in the earliest stages of CKD. In addition, the interplay between HFpEF and CKD is well-established. Please take into account the renal function parameters in the adjustments performed showing the associations of the different eicosanoids with HFpEF, both in MGH CPET and in the general population MESA with incident HFpEF.

- End-stage renal disease was defined as eGFR<30ml/min/1.73m². ESRD is a wrong term for this, at maximum advanced CKD stage (KDIGO stage G IV/V)

- Smoking is taken into account as confounder to adjust for, but alcohol use is not included. Alcohol use has been shown to be related to eicosanoids (Puri et al. J Lipid Res. 2016). And specifically in the MESA cohort, it has already been shown that alcohol intake was a strong predictor of incident HFpEF (Miller PE et al. Circulation 2015). Please add alcohol consumption to all adjustments.

- The mediation analysis is well-performed using R (mediate function). However, the mediation analysis is only cross-sectionally being performed within the MGH CPET cohort. In fact, it would be more interesting to see whether in the MESA cohort the risk factors of BMI, HTN, and DM lead to incident HFpEF over time, and whether eicosanoids mediate the relationships. Please include a causal mediation analysis with time-to-event outcome of HFpEF.

- In the current cross-sectional mediation analysis, how do CRP levels relate to the different eicosanoids in the mediation percentage between BMI, HTN, Diabetes with HFpEF? It would be good to know how much CRP levels, although being downstream marker of inflammation, mediate these associations to have a comparison with eicosanoids.

- Diuretic use is highly different between the HFpEF and no HFpEF group in MGH CPET study. Diuretics have been shown to influence the synthesis of various eicosanoids (Grose et al. Prostaglandins Leukot Med. 1986; Numabe et al. Journal of Hypertension 1989), please take into account as potential confounder.

- Some discrepancies need to be discussed. How do the authors explain that 9,10 EpOME is associated with a significant lower odd of HFpEF within the MGH CPET cohort, whereas a higher level of 9,10 EpOME in the MESA cohort has almost the highest HR for incident HF?

Reviewer #4 (Remarks to the Author):

The exploratory cohort had “physiologic evidence” of HFpEF, but natriuretic peptides and invasive haemodynamic exercise stress splitted the cohort in appimate halves with and without HFpEF. To conclude on the predictive value of additional biomarkers, the cause of dyspnea in the “no HFpEF” should be considered. Were any pulmonary explorations available? How were dyspnoeic subjects without HFpEF further examined?

The authors could consider using more advanced echocardiographic measures to support a HFpEF diagnosis at rest (e' , E/E' , LV/LA strain, sPAP).

What was the clinical indication for invasive haemodynamic exercise stress in this cohort? If HFpEF diagnosis based on rest echo and natriuretic peptides, did the CPET reclassify subjects?

Given the substantial differences in the baseline characteristics in the two groups, a matched analysis of the biomarkers would strengthen the classifier models.

A multivariate analysis over the whole cohort would respond to if eicosanoids are independently associated with HFpEF diagnosis in a dyspnoeic cohort.

Out of >900 measured metabolites, 21 were associated with increased HFpEF, of which only 5 are known eicosanoids, whereas 49 (12 named) were protective.

What is the biological significance of the PGF2 epimers/metabolites as HFpEF risk marker given their lower biological activity compared with endogenous PGF2 α , which was protective? Any specific metabolic step between PFF2 α metabolism that can explain this pathway in HFpEF protection/worsening?

The EPA and DHA-derived metabolites are of major importance, proving support to the resolution of heart failure inflammation. Were resolvins, protectins, maresin included in the lipidomic profiling, and were these mediators detectable?

Do you have data on exercise-induced aggravation of mitral regurgitation? Functional MI may be closely linked to the eicosanoid pathways, as suggested by a recent study: Hofbauer et al., Metabolomics

implicate eicosanoids in severe functional mitral regurgitation. ESC Heart Fail 2022. doi:
10.1002/ehf2.14160

In Fig 2 and Fig 5 it would be helpful to classify the non-named metabolites according their mother PUFA (LA, ALA; AA, EPA, DHA metabolome). Can all metabolites from each PUFA metabolome be grouped and analysed for predictive value for

Any HF classification for the incident diagnosis in the MESA cohort to distinguish HFpEF from HFrEF and NFmrEF?

Comments from Reviewer #1:

In the current manuscript Lau et al assessed differences in fatty acid-derived eicosanoid metabolites and eicosanoid-related metabolites in HFpEF patients and in HFpEF patients following exercise. The paper reports identifying 70 pro- and anti-inflammatory eicosanoid and eicosanoid-related metabolites associated with HFpEF status. As well, distinct eicosanoid profiles characterized contributors to exercise intolerance and potentially at as mediators associated with cardiometabolic traits and HFpEF.

1. The authors have assayed a large amount of metabolite data from MGH patients and MESA cohort study, as well from individuals following exercise. As this is the first study assessing eicosanoid and eicosanoid-related metabolites in HFpEF patients it is noteworthy and a strength of the paper. However, many of the metabolites identified were unnamed and considered 'putative' dampening the enthusiasm. Will all the data be deposited in the supplemental files?

RESPONSE: We appreciate the Reviewer's request to report data for the unnamed putative eicosanoid and related metabolites. All the primary data including both named and unnamed/putative eicosanoid metabolites are available in the **Supplemental Appendix (Supplemental Tables S1-5)**.

2. The authors discuss linking comorbidities to CV remodeling and progression of HFpEF in regards to systemic inflammation. However, as the authors did not assess markers of inflammation directly. Is this feasible? As well, what was the impact of exercise on the immune response?

RESPONSE: We thank the Reviewer for the recommendation to further assess downstream markers of inflammation and explore the impact of exercise on these markers. As suggested, we examined the correlation of the 70 HFpEF-related eicosanoid metabolites with traditional downstream markers of inflammation hsCRP and IL-6 and found relatively modest correlations (Spearman r range -0.21 to 0.17 for hsCRP and -0.15 to 0.15 for IL-6, displayed in **Reviewer Figures 1 & 2**).

Reviewer Figure 1: Correlation between eicosanoid metabolites and hsCRP

Reviewer Figure 2: Correlation between eicosanoid metabolites and IL-6

We further assessed the association of hsCRP and IL-6 with exercise intolerance and found that both hsCRP and IL-6 were associated with lower peak VO₂ displayed in **Reviewer Table 1**. We believe a significant strength and novelty of our study is the interrogation of systemic inflammation using eicosanoid profiling as eicosanoids offer a more nuanced assessment of the upstream initiation and suppression of inflammation and further mechanistic understanding of the role of inflammation in HFpEF pathogenesis beyond traditional markers of inflammation like hsCRP and IL-6.

Reviewer Table 1: Association of hsCRP and IL-6 with exercise capacity

	hsCRP				IL-6			
	Beta	SE	p-value	FDR q	Beta	SE	p-value	FDR q
% predicted VO ₂	-0.170	0.044	0.0001	0.0002	-0.139	0.044	0.002	0.002

3. Can the authors obtain information such as the medication profile of the patients? This might provide information regarding potential effects on key enzymes involved in the metabolism producing eicosanoids (COX, LOX, CYP)

RESPONSE: We thank the Reviewer for the recommendation to provide additional information regarding medication use, particularly with respect to medications that potentially affect the metabolism of eicosanoid metabolites. We have added aspirin and statin use to **Table 1** and our secondary models are now further adjusted for aspirin, statin, and diuretic use (secondary model further adjusted for aspirin, statin, and diuretics use displayed in **Supplemental Table 2**). Information on non-steroid anti-inflammatory medication use was not routinely ascertained in the MGH CPET database.

Changes to the manuscript are summarized:

Page 7, paragraph 1: “Use of medications including aspirin, statins, and diuretics was higher in participants with vs without HFpEF (aspirin: 43% vs 28%, statin: 48% vs 22%, diuretics: 41% vs 13%).”

Page 21, paragraph 1: “All models were adjusted for age, sex, and plate number in primary analyses. Secondary models further adjusted for BMI, diabetes, HTN, present smoking, prevalent myocardial infarction, statin use, aspirin use, and diuretic use.”

4. The correlational findings with 8(9)-EpETE and 12,13-diHOME is interesting and provides novel insight into potential ideas for mechanisms.

RESPONSE: We thank the Reviewer for noting this observation and agree that further investigation of the relationship between 8(9)-EpETE and 12,13-diHOME may reveal novel mechanistic insights into HFpEF pathogenesis, particularly in light of experimental data showing that 12,13-diHOME may be an important exercise-induced lipokine with important metabolic effects including skeletal muscle fatty acid uptake and oxidation, as cited in the manuscript in Sanford KI et al, Cell Metab, 2018.

Comments from Reviewer #3:

The original article by Lau and colleagues investigates the role of eicosanoid and eicosanoid-related metabolites in relation to HFpEF. In total, 70 pro- and anti-inflammatory eicosanoid and eicosanoid-related metabolites were found to be associated with HFpEF. This is an elegantly performed study, which could serve as a basis for therapeutic interventions for HFpEF. However, multiple points need to be addressed:

1. **In the baseline characteristics of both studies, MGH CPET and MESA no data on kidney function is provided. Although the authors excluded patients with renal function below 30 ml/min/1.73m² in the MESA study, data on renal function, including creatinine, eGFR, and proteinuria should be provided of both studies. With various different forms of CKD, it is known that many eicosanoids are upregulated, such as PGE₂. Also some specific forms of CKD, such as ADPKD have strongly upregulated eicosanoids. Inflammation is present even in the earliest stages of CKD. In addition, the interplay between HFpEF and CKD is well-established. Please take into account the renal function parameters in the adjustments performed showing the associations of the different eicosanoids with HFpEF, both in MGH CPET and in the general population MESA with incident HFpEF.**

RESPONSE: We appreciate the Reviewer's insightful feedback regarding the important interaction between renal function with eicosanoid metabolism and HFpEF pathophysiology. As suggested, we have now included information on baseline renal function including creatinine and eGFR to **Table 1** (proteinuria was not routinely captured in both the MGH CPET and MESA cohorts). We have also performed exploratory analyses additionally adjusting for eGFR in multivariable-adjusted models (exploratory model results displayed in **Supplemental Table 2**). This showed that 84% of eicosanoid associations with HFpEF status remained significant after further adjustment for eGFR with consistent directionality and only modest attenuation in effect sizes for eicosanoids associated with both higher and lower odds of HFpEF. Of the 31 eicosanoids that were associated with HFpEF status in multivariable adjusted analyses, 26 remained associated with HFpEF status at nominal P-values of <0.05 after further adjustment for eGFR. For example, the top hit PGF₂alpha had an OR 1.70 (95% CI 1.30-2.26) in primary analyses. After additionally adjusting for additional covariates and eGFR, this OR went to 1.67 (95% CI 1.18-2.40).

Changes to the manuscript are summarized:

Page 7, paragraph 1: "Renal function was lower in individuals with HFpEF (creatinine: 1.08 mg/dL vs 0.92 mg/dL, estimated glomerular filtration rate: 72 vs 83 mL/min/1.73m²).

Page 21, paragraph 1: "All models were adjusted for age, sex, and plate number in primary analyses. Secondary models further adjusted for BMI, diabetes, HTN, present smoking, prevalent myocardial infarction, statin use, aspirin use, and diuretic use. Exploratory models further adjusted for estimated glomerular filtration rate and alcohol use (for MESA analyses only)."

2. **End-stage renal disease was defined as eGFR<30ml/min/1.73m². ESRD is a wrong term for this, at maximum advanced CKD stage (KDIGO stage G IV/V)**

RESPONSE: We thank the Reviewer for noting this and have corrected the manuscript to read 'advanced chronic kidney disease' as suggested.

- 3. Smoking is taken into account as confounder to adjust for, but alcohol use is not included. Alcohol use has been shown to be related to eicosanoids (Puri et al. J Lipid Res. 2016). And specifically in the MESA cohort, it has already been shown that alcohol intake was a strong predictor of incident HFpEF (Miller PE et al. Circulation 2015). Please add alcohol consumption to all adjustments.**

RESPONSE: We appreciate the Reviewer's recommendation to adjust analyses for alcohol use given the known association between alcohol use and eicosanoid metabolism. While alcohol use was not routinely captured in the MGH CPET database, we have included information on heavy alcohol use for the MESA sample (defined as >7 alcohol drinks per week in women and >14 alcohol drinks per week in men) in **Supplemental Table 1**. We further adjusted our incident HF analyses in MESA for heavy alcohol consumption (results are displayed in **Supplemental Table 5**). These results show that for all 18 eicosanoid metabolites that were examined in association with incident HF, their associations did not appreciably differ after adjustment for heavy alcohol use. For example, our top named eicosanoid 11(S)HEPE was associated with a HR 0.79 (95% CI 0.69-0.90, P=0.0005) in the primary model, and after further adjustment for heavy alcohol use in the multivariable model had a HR 0.79 (95% CI 0.69-0.90, P=0.0005).

Changes to the manuscript are summarized:

Page 11, paragraph 1: "Results were similar after further adjustment for heavy alcohol consumption (Supplemental Table 5)."

Page 21, paragraph 1: "Exploratory models further adjusted for estimated glomerular filtration rate and alcohol use (for MESA analyses only)."

- 4. The mediation analysis is well-performed using R (mediate function). However, the mediation analysis is only cross-sectionally being performed within the MGH CPET cohort. In fact, it would be more interesting to see whether in the MESA cohort the risk factors of BMI, HTN, and DM lead to incident HFpEF over time, and whether eicosanoids mediate the relationships. Please include a causal mediation analysis with time-to-event outcome of HFpEF.**

RESPONSE: We thank the Reviewer for the recommendation to examine whether eicosanoids mediate the relationship between comorbidities with incident HFpEF in the MESA sample, and that this would provide potential interesting insights. As suggested, we performed additional exploratory mediation analyses using time-to-event data in MESA to examine whether eicosanoids and related metabolites mediate the association of clinical risk factors BMI, HTN, and DM with the development of HF. Given limited number of HFpEF events (67 events), we examined incident overall HF (283 events). Among the 70 eicosanoids associated with HFpEF in cross-sectional analyses, our exploratory mediation analyses identified 6 eicosanoid metabolites that may mediate the association of BMI with incident HF, 4 eicosanoid mediators of HTN and incident HF, and 4 eicosanoid mediators of DM with incident HF in MESA (p<0.05 for all). Results are displayed in **Reviewer Table 2**. We acknowledge that the small number of HFpEF events precludes further time-to-event analyses. Given this acknowledged limitation, we favor keeping cross-sectional mediation analyses in the manuscript as currently presented.

Changes to the manuscript are summarized:

Page 17, paragraph 1: "Further, given limited number of HFpEF events in the MESA sample, we were not powered to specifically examine incident HFpEF or to perform mediation analyses to examine whether eicosanoids may mediate the association of clinical risk factors with incident HFpEF."

Reviewer Table 2: Mediation analysis of eicosanoid metabolites as potential mediator of association of BMI, diabetes, and HTN with incident HF in MESA

m/z	RT	Putative ID	Mediation		Covariate → EIC			EIC → HFpEF			Direct effect			Total effect		
			Effect	p-value	β	SE	p-value	β	SE	p-value	β	SE	p-value	β	SE	p-value
BMI																
309.2075	4.6990	Unknown	0.24	<0.001	0.15	0.01	<0.001	-0.37	0.05	<0.001	-0.17	0.05	0.001	-0.23	0.05	<0.001
429.2375	2.3310	Unknown	0.11	<0.001	0.11	0.01	<0.001	-0.23	0.06	<0.001	-0.20	0.05	<0.001	-0.23	0.05	<0.001
293.2100	5.3897	EIC_12	0.12	<0.001	0.13	0.01	<0.001	-0.21	0.06	<0.001	-0.20	0.05	<0.0011	-0.23	0.05	<0.001
317.2054	4.2242	11(S) HEPE	0.08	0.03	-0.14	0.01	<0.001	0.13	0.06	0.03	-0.21	0.05	<0.001	-0.23	0.05	<0.001
357.2072	4.0515	Putative PG	0.04	0.03	-0.05	0.01	<0.001	0.17	0.08	0.02	-0.22	0.05	<0.001	-0.23	0.05	<0.001
355.2493	5.1507	Unknown	0.07	0.03	0.14	0.01	<0.001	-0.13	0.06	0.03	-0.21	0.05	<0.001	-0.23	0.05	<0.001
HTN																
309.2075	4.6990	Unknown	0.10	<0.001	0.33	0.03	<0.001	-0.32	0.05	<0.001	-1.05	0.13	<0.001	-1.17	0.13	<0.001
429.2375	2.3310	Unknown	0.03	<0.001	0.15	0.03	<0.001	-0.22	0.06	<0.001	-1.14	0.13	<0.001	-1.17	0.13	<0.001
293.2100	5.3897	EIC_12	0.05	0.01	0.32	0.03	<0.001	-0.15	0.06	0.10	-1.12	0.13	<0.001	-1.17	0.13	<0.001
323.2597	6.2037	Unknown	0.02	0.04	0.16	0.03	<0.001	-0.12	0.06	0.04	-1.15	0.13	<0.001	-1.17	0.13	<0.001
DM																
309.2075	4.6990	Unknown	0.36	<0.001	1.41	0.04	<0.001	-0.25	0.06	<0.001	-0.64	0.17	<0.001	-1.03	0.13	<0.001
429.2375	2.3310	Unknown	0.03	0.002	0.14	0.04	<0.001	-0.23	0.06	<0.001	-1.00	0.13	<0.001	-1.03	0.13	<0.001
293.2100	5.3897	EIC_12	0.08	0.006	0.48	0.04	<0.001	-0.16	0.06	0.005	-0.95	0.13	<0.001	-1.03	0.13	<0.001
323.2597	6.2037	Unknown	0.02	0.02	0.15	0.04	<0.001	-0.14	0.06	0.02	-1.01	0.13	<0.001	-1.03	0.13	<0.001
317.2054	4.224	11(S) HEPE	0.01	0.06	-0.08	0.04	0.04	0.15	0.06	0.01	-1.02	0.13	<0.001	-1.03	0.13	<0.001

5. **In the current cross-sectional mediation analysis, how do CRP levels relate to the different eicosanoids in the mediation percentage between BMI, HTN, Diabetes with HFpEF? It would be good to know how much CRP levels, although being downstream marker of inflammation, mediate these associations to have a comparison with eicosanoids.**

RESPONSE: As suggested, we have performed additional mediation analyses modeling BMI, DM, and HTN as the primary exposure, HFpEF status as the outcome, and hsCRP as the potential mediator. We found that hsCRP mediated >25% of the association of BMI, DM, and HTN with HFpEF (35% for BMI, 29% for DM, and 32% for HTN). Results are displayed in the **Reviewer Table 3** below.

Reviewer Table 3: Mediation analysis of hsCRP as potential mediator of association of BMI, diabetes, and HTN with HFpEF

Covariate	Mediation		Covariate → EIC			hsCRP → HFpEF			Direct effect			Total effect		
	Effect	p-value	β	SE	p-value	β	SE	p-value	β	SE	p-value	β	SE	p-value
BMI	0.35	<0.001	0.46	0.04	<0.001	0.50	0.11	<0.001	0.44	0.11	<0.001	0.63	0.10	<0.001
Diabetes	0.29	<0.001	0.53	0.09	<0.001	0.59	0.11	<0.001	0.76	0.20	<0.001	0.98	0.18	<0.001
HTN	0.32	<0.001	0.55	0.12	<0.001	0.64	0.11	<0.001	0.73	0.28	0.009	1.02	0.26	<0.001

6. **Diuretic use is highly different between the HFpEF and no HFpEF group in MGH CPET study. Diuretics have been shown to influence the synthesis of various eicosanoids (Grose et al. Prostaglandins Leukot Med. 1986; Numabe et al. Journal of Hypertension 1989), please take into account as potential confounder.**

RESPONSE: We appreciate the Reviewer’s recommendation to adjust for diuretic use as a potential confounder. As the Reviewer astutely notes, diuretic use differed between HFpEF vs no HFpEF groups and is known to influence synthesis of specific eicosanoid metabolites. This is challenging because diuretic use is invariably also tied to HFpEF status. However, we have updated our secondary model to further adjust for aspirin, statin, and diuretic use (secondary model results displayed in **Supplemental Table 2**).

Changes to the manuscript are summarized:

Page 7, paragraph 1: “Use of medications including aspirin, statins, and diuretics was higher in participants with vs without HFpEF (aspirin: 43% vs 28%, statin: 48% vs 22%, diuretics: 41% vs 13%).”

Page 21, paragraph 1: “All models were adjusted for age, sex, and plate number in primary analyses. Secondary models further adjusted for BMI, diabetes, HTN, present smoking, prevalent myocardial infarction, **statin use, aspirin use, and diuretic use.**”

7. **Some discrepancies need to be discussed. How do the authors explain that 9,10 EpOME is associated with a significant lower odd of HFpEF within the MGH CPET cohort, whereas a higher level of 9,10 EpOME in the MESA cohort has almost the highest HR for incident HF?**

RESPONSE: We thank the Reviewer for the opportunity to expand our discussion related to the directionally discordant associations observed for 9,10 EpOME with HFpEF status in the MGH CPET cohort and incident HF in MESA. We offer two potential hypotheses. First, we examined the association of 9,10 EpOME with incident HF subtypes (HFpEF and HFrEF) to evaluate whether the observed

positive association between 9,10 EpOME with incident HF in MESA was driven by incident HFrEF (and not HFpEF). While these analyses are limited by power (given small number of HF events in MESA), we did find a suggestive association between 9,10 EpOME and greater risk of incident HFpEF (HR 1.34, 95% CI 0.99-1.82, $p=0.06$), but no relationship between 9,10 EpOME with incident HFrEF (HR 1.08, 95% CI 0.77-1.51, $p=0.64$). These findings argue against the hypothesis that the directionally discordant observations we observed for 9,10 EpOME with HF in the MGH CPET and MESA samples can be explained by differences in HF subtypes. We propose instead that the MGH CPET and MESA samples represent complementary, yet distinct patient populations. It is biologically plausible that higher levels of 9,10 EpOME may be associated with the development of future HF in an ostensibly healthy community sample but also associated with lower odds of HFpEF status within a sample of patients with chronic dyspnea. As the causal role of 9,10 EpOME in HFpEF has not yet been established, the directionality of the relationships between 9,10 EpOME with both incident HF and HFpEF status is not known. For example, while higher levels of 9,10 EpOME may contribute to development of future HF, clinical HFpEF syndrome may result in downregulation of 9,10 EpOME. Lastly, we acknowledge that our biological understanding of eicosanoids as both potential substrates and products of interlinked enzymatic pathways is yet to be refined, and that complex relationships along biological pathways may not be captured in our analyses. We have expanded our **Discussion** as summarized:

Page 17, paragraph 1: “The different patient characteristics may explain the directionally discordant associations observed between several eicosanoid metabolites and HF in the MGH CPET and MESA analyses. For example, 9,10 EpOME was associated with lower odds of HFpEF within the MGH CPET cohort but strongly associated with incident HF in MESA. Moreover, we acknowledge that our biological understanding of eicosanoids as both potential substrates and products of interlinked enzymatic pathways is yet to be characterized, and that our analyses may not capture complex relationships along biological pathways. Fifth, given limited number...”

Comments from Reviewer #4:

- 1. The exploratory cohort had “physiologic evidence” of HFpEF, but natriuretic peptides and invasive haemodynamic exercise stress splitted the cohort in appimate halves with and without HFpEF. To conclude on the predictive value of additional biomarkers, the cause of dyspne in the “no HFpEF” should be considered. Were any pulmonary explorations available? How were dyspnoeic subjects without HFpEF further examined?**

RESPONSE: We appreciate the Reviewer’s astute observation that individuals who did not meet physiologic HFpEF criteria in the MGH CPET sample were not entirely healthy. As the Reviewer notes, individuals without HFpEF had a number of comorbid conditions, particularly pulmonary comorbidities, including COPD, ILD, and OSA. We have included information on the proportion of participants with and without HFpEF with co-existing COPD, ILD, and OSA in **Table 1**.

- 2. The authors could consider using more advanced echocardiographic measures to support a HFpEF diagnosis at rest (e’, E/E’, LV/LA strain, sPAP).**

RESPONSE: We thank the Reviewer for the suggestion to examine echocardiographic measures as further corroborative evidence of HFpEF diagnosis. Echocardiography was not routinely performed alongside the CPET in the MGH sample. We did obtain and review clinical echo reports for n=407 participants with available data within 1 year of the CPET date. We recognize this as a limitation, and have included the most widely available measures, including the presence of left atrial enlargement and left ventricular hypertrophy, for participants with and without HFpEF in **Table 1**. Prevalence of left atrial enlargement and left ventricular hypertrophy was greater among individuals with vs without HFpEF (left atrial enlargement: 41% vs 20%, left ventricular hypertrophy: 29% vs 11%). Unfortunately, other advanced echocardiographic measures including tissue Doppler, pulmonary artery pressures, and cardiac strain were not routinely ascertained in the MGH CPET cohort.

- 3. What was the clinical indication for invasive haemodynamic exercise stress in this cohort? If HFpEF diagnosis based on rest echo and natriuretic peptides, did the CPET reclassify subjects?**

RESPONSE: We thank the Reviewer for the opportunity to expand on the unique hemodynamic HFpEF definition used in our study. All individuals in the MGH CPET sample were clinically referred for invasive CPET in the setting of unexplained dyspnea. This allowed us to define HFpEF using rigorous physiologic criteria based on careful assessment of invasive hemodynamic measures at rest and with exercise. In response to the Reviewer’s comment, we examined individuals who met HFpEF criteria by non-invasive resting echocardiographic and biomarker criteria as requested (defined as NT-pro BNP >125 pg/mL and the presence of left atrial enlargement and/or left ventricular hypertrophy). This rigorous definition led to reclassification of 45% of individuals without a diagnosis of HFpEF based on resting parameters. Results are displayed in **Reviewer Table 4** below and demonstrate that the majority (93%) of individuals deemed not to have physiologic HFpEF also did not meet echo/NP criteria. By contrast, among those with physiologic HFpEF, 33% also met echo/NP criteria. Conversely, of individuals who met echo/NP criteria for HFpEF, 80% also met physiologic HFpEF criteria; of those without echo/NP criteria for HFpEF, 55% had no evidence of physiologic HFpEF. We acknowledge that the resting echocardiographic measures we employed to define resting HFpEF in our reclassification analyses are limited. Other echocardiographic measures routinely used for the diagnosis of HFpEF were not reliably captured in the MGH CPET cohort.

Reviewer Table 3: Reclassification of HFpEF based on physiologic criteria

	HFpEF by physiologic criteria	No HFpEF by physiologic criteria
HFpEF by echocardiography and NT-pro BNP	70	18
No HFpEF by echocardiography and NT-pro BNP	145	174

These data highlight existing challenges in the definition of HFpEF, and are in keeping with other studies that demonstrate a sizeable proportion of individuals with known HFpEF who may have normal natriuretic peptide levels and/or echocardiography, as cited in the manuscripts Verbrugge FH et al European Heart Journal 2022 and Shah AM et al J Am Coll Cardiol 2019.

We acknowledge the focus on hemodynamic definition of HFpEF in our manuscript in the text below:

Page 18, paragraph 2: “Second, we studied patients referred for clinically indicated level 3 CPET. While this allowed us to define HFpEF using rigorous physiologic criteria with careful assessment of exercise physiology, we acknowledge that referral bias may limit generalizability to other HFpEF samples, including HFpEF registries or clinical trials. In that context, we further note individuals in MGH CPET who did not meet physiologic HFpEF criteria are not necessarily ‘healthy’ controls.”

4. Given the substantial differences in the baseline characteristics in the two groups, a matched analysis of the biomarkers would strengthen the classifier models.

RESPONSE: We appreciate the Reviewer’s point and as suggested, performed a propensity matched analysis examining differences in eicosanoid metabolite concentrations in individuals with and without HFpEF matched for key covariates age, BMI, and HTN. We used two approaches: optimal full matching and nearest neighbor matching without replacement using Mahalanobis distance with a caliper width defined at 0.1 standard deviation units. Optimal full matching examined all participants with and without HFpEF (N=510), whereas the sample size after nearest neighbor matching was N=334. Both matching techniques resulted in adequate balancing as indicated by minimal standardized mean differences for age, BMI, and HTN in treated (HFpEF) and control (no HFpEF) subjects (<0.1 SD). Details of matching are displayed in **Reviewer Figures 3 and 4**. We found directionally consistent findings between the primary analyses and both matched analysis (displayed in **Supplemental Table 4** in attached Excel sheet).

Reviewer Figure 3: Density & Covariate Balance Plots of Age, BMI, and HTN in Unmatched and Matched Analysis Using Optimal Full Matching

Gray line represents participants without HFpEF. Black line represents participants with HFpEF.

Reviewer Figure 4: Density and Covariate Plots of Age, BMI, and HTN in Unmatched and Matched Analysis Using Nearest Neighbor Matching with a Caliper Width Defined at 0.1 Standard Deviations

Gray line represents participants without HFpEF. Black line represents participants with HFpEF.

While these results show consistency with respect to our primary analyses, we favor keeping the current analyses in the manuscript as is to avoid diminishing sample size in the matching process. We have added text to acknowledge this point as follows:

Page 18, paragraph 2: “Third, there were substantial differences in baseline characteristics between participants with and without HFpEF that may have contributed to observed associations between eicosanoid metabolites and HFpEF. To account for differences in comorbidity profiles between HFpEF and control subjects, we matched participants by key covariates age, BMI, and HTN, and found that findings were consistent with the primary analyses.”

5. A multivariate analysis over the whole cohort would respond if eicosanoids are independently associated with HFpEF diagnosis in a dyspnoeic cohort.

RESPONSE: We thank the Reviewer for the insightful suggestion to examine the association of eicosanoids with HFpEF status across the entire MGH CPET cohort with adjustment for key covariates to

fully capture the independent relationship between eicosanoid metabolites with HFpEF status. Building upon our existing multivariable adjusted model, we now examine the association of eicosanoids with HFpEF status across the MGH CPET cohort further adjusted for aspirin, statin, and diuretic use. Results were consistent albeit attenuated after accounting for these additional potential confounders (updates to **Table 2** and **Supplemental Table 2**).

Changes to the manuscript are summarized:

Page 7, paragraph 1: “Use of medications including aspirin, statins, and diuretics was higher in participants with vs without HFpEF (aspirin: 43% vs 28%, statin: 48% vs 22%, diuretics: 41% vs 13%).”

Page 21, paragraph 1: “All models were adjusted for age, sex, and plate number in primary analyses. Secondary models further adjusted for BMI, diabetes, HTN, present smoking, prevalent myocardial infarction, **statin use, aspirin use, and diuretic use.**”

- 6. Out of >900 measured metabolites, 21 were associated with increased HFpEF, of which only 5 are known eicosanoids, whereas 49 (12 named) were protective. What is the biological significance of the PGF2 epimers/metabolites as HFpEF risk marker given their lower biological activity compared with endogenous PGF2 α , which was protective? Any specific metabolic step between PFF2 α metabolism that can explain this pathway in HFpEF protection/worsening?**

RESPONSE: We thank the Reviewer for the opportunity to further expand on the biological significance of PGF2 epimers as HFpEF risk markers particularly in the context of the lower odds of HFpEF status observed for endogenous PGF2 α (15k-PGF2 α). While little is known about the unique biological properties of PGF2 α and its epimers with respect to HFpEF risk, another the 8-epimer of PGF2 α is an established marker of oxidative stress and may explain the association between 15R-PGF2 α and 11 β -dihk-PGF2 α with greater odds of HFpEF. We have further expanded our **Discussion** as summarized:

Page 13, paragraph 2 to page 14, paragraph 1: In this context, the associations of 15R-PGF2 α and 11 β -dihk-PGF2 α with HFpEF are notable given the established role of HTN in HFpEF pathobiology and female susceptibility to HFpEF, **though interestingly the endogenous 15k-PGF2 α metabolite was associated with lower odds of HFpEF. Little is known about the distinct biological activities of endogenous PGF2 α vs its epimers particularly in the context of HFpEF risk, but the established role of another PGFa epimer 8-iso-PGF2 α as a marker of oxidative stress highlights the unique biological properties of PGF2 α epimers that may explain the discordant associations observed between endogenous PGF2 α and PGF2 α epimers with HFpEF status. By contrast, epoxide 8(9)-EpETE demonstrated the greatest negative association with HFpEF. Epoxy-eicosatetraneic acids (EpETE).”**

- 7. The EPA and DHA-derived metabolites are of major importance, proving support to the resolution of heart failure inflammation. Were resolvins, protectins, maresin included in the lipidomic profiling, and were these mediators detectable?**

RESPONSE: We agree with the Reviewer that identification of EPA and DHA-derived metabolites involved in resolution of inflammation including resolvins, protectins, and maresin associated with lower HFpEF risk provides important support for the key mechanistic role in HFpEF pathogenesis and highlights important potential targets for HF interventions. In our primary analysis, we identified

Resolvin D1 as one key resolvin that was associated with lower odds of HFpEF in the primary analysis. We have expanded our discussion on this resolvin to highlight its importance as noted by the Reviewer. Further, while we did not identify other named protective protectins or maresin metabolites, the identities of numerous other protective eicosanoid metabolites are not presently known – these molecules are of particular interest as noted by the Reviewer, and will necessitate further chemical experiments to speciate and definitively describe in future studies.

Changes to the manuscript are summarized:

Page 8, paragraph 1: “Resolvin D1, a member of the resolvin family of metabolites involved in resolution of inflammation, was also associated with lower odds of HFpEF (OR 0.74, 95% CI 0.59-0.91).”

Page 14, paragraph 1: “We also identified resolvin D1 as a key eicosanoid metabolite associated with lower odds of HFpEF. Resolvins are important EPA and DHA derivatives that promote the resolution of inflammation. Resolvin D1 specifically has been shown to exert resolution of post-myocardial infarction inflammatory injury and delayed the onset of ventricular dysfunction and heart failure in a mouse model of myocardial infarction. The association of resolvin D1 with lower odds of HFpEF provides further support for the inflammatory basis of HFpEF pathogenesis and highlights resolvin D1 as a potential target for future HF intervention.”

- 8. Do you have data on exercise-induced aggravation of mitral regurgitation? Functional MI may be closely linked to the eicosanoid pathways, as suggested by a recent study: Hofbauer et al., Metabolomics implicate eicosanoids in severe functional mitral regurgitation. ESC Heart Fail 2022. Doi: 10.1002/ehf2.14160**

RESPONSE: We appreciate the Reviewer’s astute recommendation to examine exercise-induced mitral regurgitation. Unfortunately, information on functional mitral regurgitation was not included in the MGH CPET sample given that concurrent echocardiography was not performed but is an important area for future study.

- 9. In Fig 2 and Fig 5 it would be helpful to classify the non-named metabolites according their mother PUFA (LA, ALA; AA, EPA, DHA metabolome). Can all metabolites from each PUFA metabolome be grouped and analysed for predictive value for**

RESPONSE: We thank the Reviewer for the recommendation to further classify the non-named metabolites according to their parent PUFA. As requested, we have included putative parent PUFAs for non-named metabolites when details on their PUFA classification were available. Of note, many of the metabolites examined include novel eicosanoid and eicosanoid-related metabolites for which the identities of their parent PUFA are not known. Additional classification of metabolites by parent PUFA is now listed in **Figure 5** and **Supplemental Appendix**.

- 10. Any HF classification for the incident diagnosis in the MESA cohort to distinguish HFpEF from HFrEF and NFmrEF?**

RESPONSE: Given the limited number of incident HF events in the MESA sample (283 total HF events, of which 76 HFrEF events, and 67 HFpEF events could be classified based on available cardiac imaging studies at or around the time of HF presentation), we examined the association of eicosanoid metabolites

with incident overall HF in our primary replication analyses. We appreciate the Reviewer's request to further examine the association of eicosanoid metabolites with incident HF subtypes and acknowledge limited power to do so. We conducted exploratory analyses by HF subtype as requested and found that 1 unnamed eicosanoid metabolite was associated with incident HF_rEF and 4 (1 named [11(S)-HEPE], 3 unnamed) with incident HF_pEF (HF_mrEF was not available in MESA).

Changes to the manuscript are summarized:

Page 24, paragraph 3: “Finally, to further explore the clinical relevance of eicosanoid and related metabolites identified in primary analyses, we took eicosanoid and related metabolites associated with HF_pEF status in MGH CPET and examined their association with incident HF and HF subtypes using Cox proportional hazards models in the MESA sample **accounting for competing risks of death, other HF subtype, and unclassified HF.**”

Page 12, paragraph 2: “With respect to HF subtypes, we observed 76 HF_rEF events and 67 HF_pEF events over a median follow-up time of 17 years. We found that 1 unnamed HF_pEF-related eicosanoid metabolite was associated with incident HF_rEF and 4 metabolites (1 named, [11(S) HEPE], 3 unnamed) were associated with incident HF_pEF.”

Page 24, paragraph 4: “Finally, to further explore the clinical relevance of eicosanoid and related metabolites identified in primary analyses, we took eicosanoid and related metabolites associated with HF_pEF status in MGH CPET and examined their association with incident HF **and HF subtypes** using Cox proportional hazards models in the MESA sample.”

REVIEWERS' COMMENTS

Reviewer #1 (Remarks to the Author):

The authors have responded well to the previous concerns and updated the manuscript accordingly. There are no further concerns.

Reviewer #3 (Remarks to the Author):

The authors adequately addressed my previous comments, and the manuscript has further improved. No more comments.

Reviewer #4 (Remarks to the Author):

The authors have responded to this reviewer's comments.